# Beyond Random: Automatic Inner-loop Optimization in Dataset Distillation

**Muquan Li[1], Hang Gou[1], Dongyang Zhang[1,*], Shuang Liang[1],**
**Xiurui Xie[1], Deqiang Ouyang[2], Ke Qin[1]**
[1]University of Electronic Science and Technology of China    [2]Chongqing University
{muquanli2023, gouhang}@std.uestc.edu.cn
{dyzhang, shuangliang, xiexiurui, qinke}@uestc.edu.cn
deqiangouyang@cqu.edu.cn

## Abstract

The growing demand for efficient deep learning has positioned dataset distillation as a pivotal technique for compressing training dataset while preserving model performance. However, existing inner-loop optimization methods for dataset distillation typically rely on random truncation strategies, which lack flexibility and often yield suboptimal results. In this work, we observe that neural networks exhibit distinct learning dynamics across different training stages—early, middle, and late—making random truncation ineffective. To address this limitation, we propose Automatic Truncated Backpropagation Through Time (AT-BPTT), a novel framework that dynamically adapts both truncation positions and window sizes according to intrinsic gradient behavior. AT-BPTT introduces three key components: (1) a probabilistic mechanism for stage-aware timestep selection, (2) an adaptive window sizing strategy based on gradient variation, and (3) a low-rank Hessian approximation to reduce computational overhead. Extensive experiments on CIFAR-10, CIFAR-100, Tiny-ImageNet, and ImageNet-1K show that AT-BPTT achieves state-of-the-art performance, improving accuracy by an average of 6.16% over baseline methods. Moreover, our approach accelerates inner-loop optimization by 3.9 × while saving 63% memory cost.

## 1   Introduction

The unprecedented success of deep learning [14, 18] has been fundamentally driven by the availability of large-scale datasets and computational resources. However, this data-centric paradigm poses scalability challenges, particularly in storage efficiency, computational overhead, and environmental sustainability [5, 33, 20]. These limitations have catalyzed the emergence of dataset distillation [45, 53, 10, 48, 19] (DD) as a transformative approach to data-efficient learning. By synthesizing compact surrogate datasets that preserve the essential characteristics of their original counterparts, DD enables efficient model training without compromising performance.

Dataset distillation is inherently formulated as a bilevel optimization problem [36, 9], consisting of inner-loop and outer-loop optimization. The outer-loop optimizes the synthetic dataset to minimize discrepancies between models trained on distilled and original data, and the inner-loop simulates the training dynamics of a neural network on the distilled dataset [9]. Previous DD methods focus on approximating the outer-loop to directly align the final performance metric [53, 51, 24, 2, 6, 17, 12, 22]. However, outer-loop methods typically rely on surrogate objectives or heuristic measures to indirectly capture the effects of inner-loop training, thereby failing to accurately reflect the model's training dynamics. Popular inner-loop optimization techniques, such as Backpropagation Through

---

*Corresponding to: Dongyang Zhang

39th Conference on Neural Information Processing Systems (NeurIPS 2025).

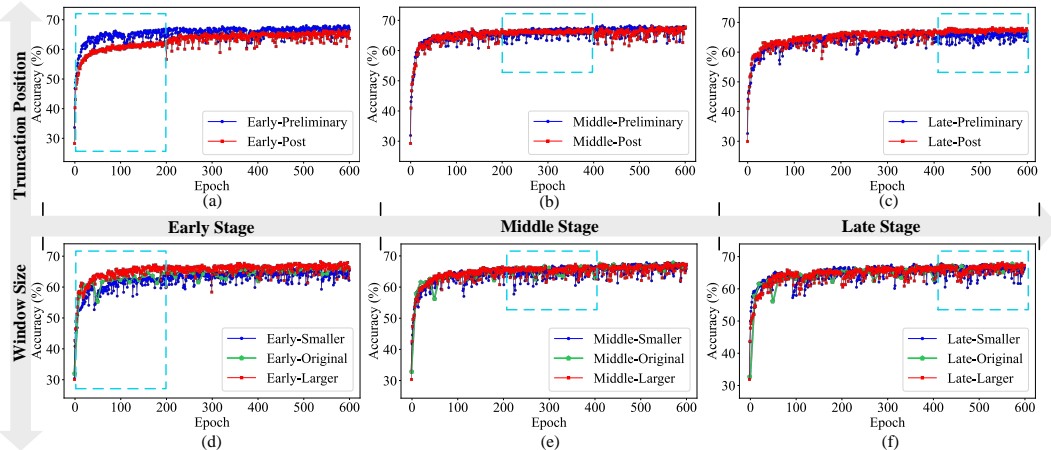

Figure 1: Hypothesis verification for the influence of truncation strategies and window size. (a)(b)(c) show experiments where the preliminary or post truncation positions are implemented at early, middle and late stages, respectively, and (d)(e)(f) present experiments where the window size is changed after fixing the truncation position. For example, Early-Preliminary in (a) means that randomly select preliminary phase (0-100) timesteps in early training stage (0-200 epochs).

Time (BPTT) [46] and its truncated variants (T-BPTT [31], RaT-BPTT [9]), adopt rigid truncation strategies to partially address performance bottlenecks. Nevertheless, these methods uniformly apply fixed or random truncation across all training stages, disregarding the distinct learning patterns of neural networks: early stages prioritize simple patterns and later stages refine complex features [1]. Consequently, the rigid truncation of learning trajectories limits model performance.

To address these limitations, we posit that automatic adjustment of truncation strategies across distinct training stages could better align with the intrinsic learning dynamics of deep neural networks (DNNs) [1]. We verify this perspective through a controlled variable experiment, which partitions the training process into three stages, and selects preliminary or post timesteps unfolding in each epoch as the truncation position. Our experimental analysis reveals three critical findings: (1) early-stage truncation of preliminary timesteps yields a 2.9% improvement in test accuracy (Fig. 1a), (2) middle-stage performance demonstrates negligible sensitivity to truncation position ($\Delta < 0.3\%$, Fig. 1b), and (3) late-stage prioritization of post timesteps enhances accuracy by 1.8% (Fig. 1c). These results substantiate the suboptimality of random truncation approaches, and motivate the formulation of a secondary perspective: adaptive window sizing could simultaneously optimize computational efficiency and gradient preservation. Subsequent experiments with fixed optimal truncation positions (Fig. 1d-f) demonstrate that a larger window in the early stage achieves a 2.5% accuracy gain, while the middle- and late-stage variations show insignificant impacts ($\Delta < 0.2\%$). These hypotheses and their validation are detailed in Section 4.1.

Based on the above observation, we propose an effective DD method called Automatic Truncated Backpropagation Through Time (AT-BPTT). This method integrates three key mechanisms: dynamic truncation position, adaptive window size and low-rank Hessian approximation. Dynamic truncation position employs gradient magnitudes to probabilistically determine better truncation timesteps across different training stages. Adaptive window size leverages the magnitudes of gradient variation to modulate the truncation window, ensuring that critical gradient information is retained when fluctuations are high. Low-rank Hessian approximation addresses the computational time and storage cost issues of Hessian matrix calculation in second-order optimization. Extensive experiments illustrate that AT-BPTT achieves state-of-the-art (SOTA) performance on CIFAR-10 [15], CIFAR-100 [15], Tiny-ImageNet [16] and ImageNet-1K [29], outperforming the leading inner-loop method by an average of 6.16% while delivering a 3.9 × faster training speed and a 63% memory reduction.

## 2 Related Work

Dataset distillation is proposed with the aim of generating a compact synthetic dataset that effectively substitutes for the original dataset. Dataset distillation can be formulated as a bilevel optimization problem, with current mainstream approaches broadly categorized into outer-loop optimization

methods (data matching methods) and inner-loop optimization methods (meta-learning methods). Outer-loop optimization methods achieve dataset distillation by aligning surrogate objectives between the original and synthetic datasets. Gradient matching [51] aligns the gradients of synthetic and original data during training to mimic the optimization behavior of the original dataset. Trajectory matching [2, 6, 12, 22, 4] aligns parameter trajectories throughout the training process, overcoming the short-term matching limitations of gradient matching. Distribution matching [41, 52, 44] aligns distributions in the feature space, circumventing the complexity of bilevel optimization. However, outer-loop methods rely on surrogate models to indirectly capture the effects of inner-loop training, failing to accurately reflect the true training dynamics of the model.

Inner-loop optimization methods treat the synthetic dataset as hyperparameters, minimizing the risk of models trained on synthetic data on the target dataset. A prominent inner-loop approach is Backpropagation Through Time [46], which simulates the model training process by performing multi-step gradient descent in the inner loop and optimizes the synthetic data in the outer loop to achieve dataset distillation. Subsequently, T-BPTT [31] improves distillation performance by truncating the unrolled time steps. Recently, RaT-BPTT [9] employs a random truncation strategy to enhance BPTT's distillation performance, partially reducing computational costs. However, its random truncation strategy fails to align with the learning characteristics of deep neural networks, limiting its performance.

Besides optimization-based techniques, there are several other DD methods. Diffusion-based methods [35, 3] generate compact samples that capture key features of the original dataset by optimizing representations in the latent space or directly adjusting the diffusion process, excelling in high-quality image generation and high-resolution tasks. Decoupled optimization methods [47, 32, 38] decompose the complex distillation process into independently optimizable subtasks, reducing computational complexity and enhancing the capability to handle large-scale datasets.

## 3 Preliminary

Dataset distillation [9] aims to compress a large training set $\mathcal{D}$ into a smaller set $\mathcal{S}$ such that training on $\mathcal{S}$ achieves comparable performance to training on $\mathcal{D}$. The process involves a bilevel formulation:

$$\min_{\mathcal{S}} \mathcal{L}(\theta_T(\mathcal{S}), \mathcal{D}) \quad \text{s.t.} \quad \theta_T(\mathcal{S}) = \mathcal{A}(\theta_0, \mathcal{S}, T), \tag{1}$$

where $\mathcal{A}$ represents the inner-loop learner over $T$ steps with initialization $\theta_0$ and synthetic dataset $\mathcal{S}$.

**Backpropagation Through Time (BPTT).** BPTT [46] is the standard method for solving bilevel optimization in reverse mode. When $\mathcal{A}$ follows gradient descent with learning rate $\alpha$, the meta-gradient with respect to every unrolling learning trajectory is computed using the chain rule:

$$G_{BPTT} = -\alpha \frac{\partial \mathcal{L}(\theta_T(\mathcal{S}), \mathcal{D})}{\partial \theta} \sum_{i=1}^{T-1} \prod_{j=i+1}^{T-1} [1 - \alpha H_j] g_i, \tag{2}$$

where $H_j = \frac{\partial^2 \mathcal{L}(\theta_j(\mathcal{S}), \mathcal{D})}{\partial \theta^2}$ represents the Hessian matrix at timestep $j$, and $g_i = \frac{\partial \mathcal{L}(\theta_i(\mathcal{S}), \mathcal{D})}{\partial \theta \partial \mathcal{S}}$ denotes the value with respect to the parameter $\theta$ and the mixed partial derivatives of $\mathcal{S}$ at timestep $i$. The detailed derivation of the formula is shown in the Appendix A.

**Truncated BPTT (T-BPTT).** To alleviate the memory burden, T-BPTT [31] propagates gradients through a smaller unrolling window of $M$ steps instead of the full trajectory:

$$G_{T-BPTT} = -\alpha \frac{\partial \mathcal{L}(\theta_T(\mathcal{S}), \mathcal{D})}{\partial \theta} \sum_{i=T-M}^{T-1} \prod_{j=i+1}^{T-1} [1 - \alpha H_j] g_i. \tag{3}$$

This truncation omits the first $T - M + 1$ terms, reducing the number of Hessian products.

**Random Truncated BPTT (RaT-BPTT).** RaT-BPTT [9] extends T-BPTT by randomly positioning the truncated window along every unrolling learning trajectory in the training process. The meta-gradient of RaT-BPTT with random truncation at position $N$ is:

$$G_{RaT-BPTT} = -\alpha \frac{\partial \mathcal{L}(\theta_N(\mathcal{S}), \mathcal{D})}{\partial \theta} \sum_{i=N-M}^{N-1} \prod_{j=i+1}^{T-1} [1 - \alpha H_j] g_i, \tag{4}$$

which differs from T-BPTT by randomly sampling $M$ timesteps and omitting shared Hessian products.

## 4 Methodology

### 4.1 Hypothesis Verification

We challenge the practice of applying random truncation uniformly across the entire training process in RaT-BPTT [9] based on two key observations. First, it is well established that deep neural networks (DNNs) tend to learn simple patterns during the early training stages before progressively acquiring more complex patterns [1]. Second, T-BPTT [31] exclusively utilizes the last $T - M + 1$ timesteps of each epoch, while the impact of restricting updates to early timesteps remains unexplored. These observations motivate our **Hypothesis I**: *Might sampling specific timesteps from distinct stages lead to better performance, rather than uniformly applying random truncation?*

To verify this hypothesis, we partition the epochs throughout the training process into three equal stages (early, middle, and late stages) while bisecting the timesteps within the unfolding learning trajectory for every epoch (preliminary and post phases). Our validation involves controlled experiments where we randomly truncate either preliminary or post timesteps during the early stage, maintaining standard RaT-BPTT implementation in subsequent two stages. Similar validations are implemented on the middle and late training stages. The experiments are conducted on the CIFAR-10 [15] dataset using the ConvNets architecture. Our experimental results shown in Fig. 1a-c reveal three key observations: (1) preliminary phase truncation in early stage enhances validation accuracy by an average of 2.9%; (2) middle stage shows negligible performance variance ($\Delta < 0.5\%$) between phase choices; (3) post phase selection in late stage demonstrates 1.8% accuracy improvement. These findings not only validate our initial hypothesis but also raise a new **Hypothesis II**: *Can dynamically adjusting the truncation window size further enhance distillation performance?*

To further investigate the impact of truncation window size, we adopt a roughly adaptive truncation strategy based on the experiments mentioned above: selecting preliminary timesteps during the early training stage, applying random truncation in the middle stage, and seclecting post timesteps in the late stage. Subsequently, we conduct experiments in which each stage alternated between using the original window (used in RaT-BPTT), a larger window, and a smaller window (modifying the original window size by ±10 timesteps). As shown in Fig. 1d–f, large window leads to a 2.5% accuracy improvement in the early stage, while variations in window size during the middle and late stages have a negligible effect ($\Delta < 0.2\%$). Based on these, we conclude that selecting appropriately sized truncation windows for specific timesteps at different training stages yields better performance.

### 4.2 Automatic Truncated BPTT

Building on the above findings, we propose an Automatic Truncated Backpropagation Through Time (AT-BPTT) framework designed to optimize the inner-loop for dataset distillation. This automatic mechanism consists of three components: dynamic truncation position, adaptive window size and a low-rank Hessian approximation, as shown in Fig. 3.

**Dynamic Truncation Position.** The experimental validation of **Hypothesis I** raises a question: How to establish an evaluation metric that enables staged truncation of timesteps during different training stages? Our analysis begins with the gradient accumulation mechanism expressed in Eqn. 4, which reveals that parameter updates represent cumulative contributions from all timestep gradients, with each timestep's influence modulated by its temporal position and subsequent gradient dynamics. To quantify these temporal effects, we record gradient magnitudes across all timesteps during training iterations, with averages visualized in Fig. 2. It is observed that gradient updates are initially large but gradually diminish as timesteps progress. Interestingly, this observation and the verification of **Hypothesis I** align with Arpit et al. [1]'s seminal work on deep learning dynamics: DNNs tend to learn simple, easily identifiable patterns during

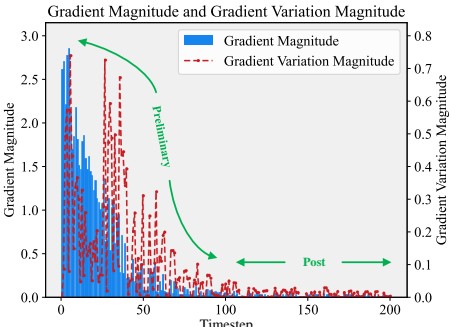

Figure 2: Illustration of the gradient and gradient variation average magnitudes each timestep during training process. The entire timesteps are roughly averaged into preliminary and post phases.

the early stages of training before progressively shifting toward more complex and fine-grained patterns in later stages. This is attributable to that simple patterns often dominate the data and exhibit

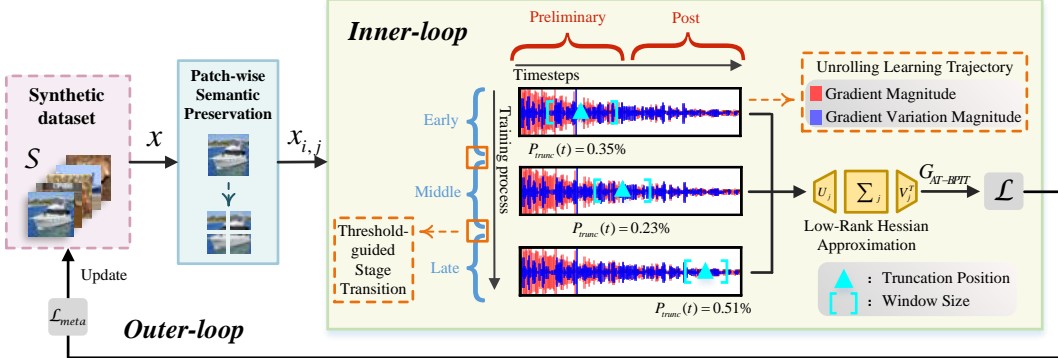

Figure 3: Overall framework of our proposed AT-BPTT. The distilled data flows through our patch-wise semantic preservation module in the inner-loop optimization. The dynamic truncation position and adaptive window size then jointly optimize inner-loop training dynamics. The low-rank Hessian approximation is utilized to reduce computational cost.

high saliency, allowing large gradient updates in the preliminary phase to align effectively with these patterns, leading to rapid performance improvement in the early stage. Although the gradients of timesteps in the post phase are smaller, they play a crucial role in fine-tuning the network, ultimately leading to significant performance gains in the late stage.

This staged behavior suggests the gradient magnitude serves as an intrinsic metric for training stage identification. We thus formalize our truncation probability mechanism through temperature-controlled softmax normalization:

$$P_{\text{trunc}}(t) = \frac{\exp(\|\nabla_\theta \mathcal{L}_t\|_2/\tau)}{\sum_{i=1}^T \exp(\|\nabla_\theta \mathcal{L}_i\|_2/\tau)}, \tag{5}$$

where $\tau$ represents the annealing temperature parameter. Therefore, we define a three-stage strategy to determine dynamic truncation positions: (1) In the early stage, each timestep at $t$ is selected as a truncation position with probability $P_{\text{trunc}}(t)$; (2) In the middle stage, truncation is applied randomly; (3) In the late stage, the probability of each timestep is $\frac{1-P_{\text{trunc}}(t)}{T-1}$:

$$P_{\text{trunc}}(t) = \begin{cases} \frac{\exp(\|\nabla_\theta \mathcal{L}_t\|_2/\tau)}{\sum_{i=1}^T \exp(\|\nabla_\theta \mathcal{L}_i\|_2/\tau)}, & \text{if Early Stage,} \\ 1/T, & \text{if Middle Stage,} \\ \frac{1-\frac{\exp(\|\nabla_\theta \mathcal{L}_t\|_2/\tau)}{\sum_{i=1}^T \exp(\|\nabla_\theta \mathcal{L}_i\|_2/\tau)}}{T-1}, & \text{if Late Stage.} \end{cases} \tag{6}$$

**Adaptive Window Size.** Based on the three-stage framework, we further propose an adaptive window size mechanism to address **Hypothesis II**. Revisiting Eqn.4, conventional inner-loop optimizations employ fixed window size truncation to reduce computational overhead by partial Hessian matrix computation. This Hessian matrix, obtained via second-order differentiation of the loss function, captures the temporal variations in gradients across different timesteps. We thus record the gradient variation magnitude $|\|\nabla_\theta L_t\|_2 - \|\nabla_\theta L_{t-1}\|_2|$ at each timestep during training as visualized in Fig. 3. This reveals a type of staged pattern: early training stages exhibit high-magnitude oscillatory gradients, while middle and late stages demonstrate stabilized gradient variations.

This observation aligns with **Hypothesis II** verification experiments, suggesting that substantial gradient variations indicate active adjustment of model decision boundaries at corresponding timesteps. To obtain an adaptive window size aligned with gradient variation magnitude, we introduce a temperature-controlled soft normalization of gradient variations to obtain the size weight $\eta(t)$ at $t$:

$$\eta(t) = \frac{\exp(|\|\nabla_\theta \mathcal{L}_t\|_2 - \|\nabla_\theta \mathcal{L}_{t-1}\|_2|/\tau)}{\sum_{i=1}^T \exp\left(|\|\nabla_\theta \mathcal{L}_i\|_2 - \|\nabla_\theta \mathcal{L}_{i-1}\|_2|/\tau\right)}. \tag{7}$$

The adaptive window size $W^*(t)$ at timestep $t$ is then computed through linear transformation of the original window size $W$ in RaT-BPTT: $W^*(t) = W - d + 2d \cdot \eta(t)$ with a range of $[W - d, W + d]$ and $d \gg 0$. This adaptive adjustment enables the model to automatically allocate larger windows for

timesteps with significant gradient variations while contracting windows for stable phases, achieving better balance between computational efficiency and model performance.

**Threshold-guided Stage Transition.** In contrast to the rough equal partitioning for hypothesis verification in Section 4.1, we adopt a threshold-guided stage transition approach to stably switch between early, middle, and late training stages. Since the convergence of optimization processes in visual tasks is ultimately determined by the trend in model performance metrics, we utilize accuracy variation at each timestep to automatically regulate training stages. We define the accuracy variation as: $\Delta A_t = A_t - A_{t-1}$. Two zero-initialized counters $C_{early-middle}$ and $C_{middle-late}$ are defined to record the number of times $\Delta A_t < M$ and $\Delta A_t < N$, respectively. The rule from early to middle stage and from middle to late stage are:

$$C_{early-middle} = \sum_{t=1}^{T_1} \mathbf{1}(\Delta A_t < M) \geq X, \quad C_{middle-late} = \sum_{t=T_1+1}^{T_2} \mathbf{1}(\Delta A_t < N) \geq Y, \quad (8)$$

where $\mathbf{1}(\cdot)$ denotes the indicator function that returns 1 if the condition is satisfied and 0 otherwise. This mechanism triggers stage transition when either: (1) $X$ times of accuracy variation fall below $M$ for early-to-middle transition, or (2) $Y$ times fall below $N$ for middle-to-late transition. The accuracy variation reflects model learning dynamics, and employing multiple threshold evaluations effectively mitigates training noise, thereby ensuring stable transitions.

## 4.3 Low-Rank Hessian Approximation

In addition to random truncation, RaT-BPTT's unfolding learning trajectory approach also suffers from high computational overhead. This primarily stems from the frequent computation of implicit Hessian matrix products in Eqn. 4, which are required for evaluating inner-loop optimization performance. Therefore, we introduce a low-rank Hessian approximation (LRHA) to reduce both time and memory complexity while preserving the gradient direction by leveraging the low-rank structure of the Hessian matrix. Given the Hessian at timestep $j$, we approximate it by a rank-$k_j$ factorization:

$$\tilde{H}_j = U_j \Sigma_j V_j^\top \approx H_j, \tag{9}$$

where $U_j, V_j \in \mathbb{R}^{d \times k_j}$ and $\Sigma_j \in \mathbb{R}^{k_j \times k_j}$. We adaptively determine $k_j$ based on the normalized gradient magnitude, with a lower bound:

$$k_j = \max\left(k_{\min}, \left\lfloor k_{\max} \cdot \frac{\|\nabla_\theta \mathcal{L}_j\|_2}{\max_{i \leq j} \|\nabla_\theta \mathcal{L}_i\|_2} \right\rfloor \right), \tag{10}$$

where $k_{\max} = 0.1\,d$ and $k_{\min} > 0$ ensures numerical stability. Instead of materializing $H_j$, we apply randomized singular value decomposition (SVD) using Hessian-vector products (HVP) and QR factorization:

$$Y^{(0)} = \text{HVP}(H_j, \Omega_j), \quad \Omega_j \sim \mathcal{N}(0, I),$$
$$Y^{(q)} = \text{HVP}\left(H_j, H_j^\top Y^{(q-1)}\right), \quad q = 1, 2, \tag{11}$$
$$Q_j, R_j = \text{QR}(Y^{(2)}), \quad B_j = Q_j^\top H_j Q_j, \quad \tilde{U}_j, \Sigma_j, V_j = \text{SVD}(B_j).$$

We then reconstruct new $\tilde{H}_j$:

$$\tilde{H}_j = (Q_j \tilde{U}_j) \cdot \Sigma_j \cdot (Q_j V_j)^\top. \tag{12}$$

The final meta-gradient of AT-BPTT is formulated as:

$$G_{\text{AT-BPTT}} = -\alpha \frac{\partial \mathcal{L}(\theta_N(\mathcal{S}), \mathcal{D})}{\partial \theta} = \sum_{i=N-W^*(t)}^{N-1} \prod_{j=i+1}^{T-1} [1 - \alpha \tilde{H}_j] \cdot P_{\text{trunc}}(i) g_i. \tag{13}$$

Let $p$ denote the dimensionality of the model parameter vector. Through our proposed LRHA, the time and memory complexity drop from $\mathcal{O}(p^2)$ to $\mathcal{O}(p\,k_j + k_j^3)$ and from $\mathcal{O}(p^2)$ to $\mathcal{O}(2pk_j + k_j^2)$, respectively, greatly reducing computational resource consumption.

### 4.4 Patch-wise Semantic Preservation

To address the poor performance of inner-loop DD on high-resolution datasets, we propose a lightweight patch-wise semantic preservation (PSP) to enhance generalization capability. Given an input image $x \in \mathbb{R}^{H \times W \times 3}$, we split it into $n \times n$ non-overlapping patches:

$$\mathcal{P} = \{x_{ij} | x_{ij} = x[i \cdot s : (i+1) \cdot s, j \cdot s : (j+1) \cdot s], \quad s = \lfloor H/n \rfloor\} \tag{14}$$

where $n$ controls the granularity of local processing and we set $n = 4$. Each patch $x_{ij}$ is independently distilled using AT-BPTT to generate local synthetic prototypes:

$$\mathcal{S}_{ij} = \text{AT-BPTT}(x_{ij}, \theta_{\text{local}}), \tag{15}$$

where $\theta_{\text{local}}$ denotes the parameters of the local distillation network applied to each patch. To ensure semantic coherence, we perform prototype centroid matching against the global prototype set $\mathcal{S}_{\text{global}} = \{\mathcal{S}_{ij}\}_{i,j=1}^{n}$:

$$\mathcal{L}_{\text{align}} = \sum_{i,j} \|\mu(\mathcal{S}_{ij}) - \mu(\mathcal{S}_{\text{global}})\|_2, \tag{16}$$

where $\mu(\cdot)$ calculates the prototype centroid. The final objective combines original distillation loss and alignment loss: $\mathcal{L}_{\text{total}} = \mathcal{L}_{\text{AT-BPTT}} + \lambda \mathcal{L}_{\text{align}}$, where $\lambda$ balances the two objectives.

## 5 Experiments

### 5.1 Experimental Setup

**Dataset.** We adhere to the conventional procedure adopted in dataset distillation [9]. We select three standard datasets: CIFAR-10 [15] (10 classes, 32×32), CIFAR-100 [15] (100 classes, 32×32), and Tiny-ImageNet [16] (200 classes, 64×64). To further show the effectiveness of AT-BPTT on high-resolution images, we scale up the dataset to ImageNet-1K [29] (1,000 classes, 224×224). For CIFAR-10 and CIFAR-100, we distill datasets with 1, 10, and 50 images per class (IPC = 1, 10, 50), while for Tiny-ImageNet and ImageNet-1K, we use 1 and 10 images per class (IPC = 1, 10).

**Baselines.** We compare our AT-BPTT with the most representative baselines categorized into two distinct paradigms: one category contains methods that approximate or optimize the outer-loop of bilevel optimization, and the other category includes methods that directly improve the inner-loop of bilevel optimization. More details on baseline methods are introduced in Appendix B.

**Implementation Details.** Following Rat-BPTT [9], we employ standardized convolutional neural network [11] (CNN) architectures with depth adapted to dataset specifications: a 3-layer CNN (Conv-3) for CIFAR-10/CIFAR-100 synthetic datasets, and a 4-layer CNN (Conv-4) for Tiny-ImageNet and high-resolution ImageNet-1K datasets. More hyperparameter settings are detailed in Appendix B.

### 5.2 Comparison with Previous Methods

Tab. 1 demonstrates that the proposed AT-BPTT framework establishes new state-of-the-art performance on three benchmark datasets (CIFAR-10 [15], CIFAR-100 [15], and Tiny-ImageNet [16]), outperforming existing inner-loop and outer-loop optimization approaches by a significant margin. Most remarkably, AT-BPTT achieves an average accuracy gain of 6.16% over RaT-BPTT, the SOTA inner-loop DD method. This substantial improvement in low-resolution image distillation stems from the combination of our novel dynamic truncation strategy and adaptive window size mechanism, which collectively enhance the model's capacity to capture multi-level feature correlations through dynamically adjusted learning trajectories. The visualizations of these synthetic datasets are shown in Appendix F.

For the high-resolution dataset ImageNet-1K, Tab. 1 shows that AT-BPTT also exhibits superior competitiveness compared to leading methods, achieving 30.6% accuracy under the IPC=10 setting. This exceptional performance is attributed to the transformative role of PSP segmentation strategy, which effectively transfers AT-BPTT's strengths in low-resolution domains to high-resolution scenarios. These results not only confirm AT-BPTT's capability in processing high-resolution data but also highlight the scalability of the PSP strategy for more advanced high-resolution applications. More experimental results are provided in Appendix C.

Table 1: Comparison with the SOTA baseline dataset distillation methods. Following previous methods, the ConvNet used for distillation are Conv-3 on CIFAR-10 [15] and CIFAR-100 [15], Conv-4 on Tiny-ImageNet [16] and ImageNet-1K [29]. Each reported result is the average of 5 experiments. Entries with "-" are absent due to scalability problems.

| Dataset | CIFAR-10 | | | CIFAR-100 | | | Tiny-ImageNet | | ImageNet-1K | |
|---|---|---|---|---|---|---|---|---|---|---|
| Img/class(IPC) | 1 | 10 | 50 | 1 | 10 | 50 | 1 | 10 | 1 | 10 |
| *Outer-loop Optimization* | | | | | | | | | | |
| DSA [53] | 28.8±0.7 | 52.1±0.5 | 60.6±0.5 | 13.9±0.3 | 32.3±0.3 | 42.8±0.4 | 6.6±0.2 | 14.4±2.0 | 1.1±0.7 | 3.2±0.3 |
| CAFE [41] | 30.3±1.1 | 46.3±0.6 | 55.5±0.6 | 12.9±0.3 | 27.8±0.3 | 37.9±0.3 | - | - | - | - |
| MTT [2] | 46.2±0.8 | 65.4±0.7 | 71.6±0.2 | 24.3±0.3 | 39.7±0.4 | 47.7±0.2 | 8.8±0.3 | 23.2±0.2 | - | - |
| TESLA [6] | 48.5±0.8 | 66.4±0.8 | 72.6±0.7 | 24.8±0.4 | 41.7±0.3 | 47.9±0.3 | - | - | 7.7±1.0 | 17.8±0.5 |
| DATM [12] | 46.9±0.5 | 66.8±0.2 | 76.1±0.3 | 27.9±0.2 | 47.2±0.4 | 55.0±0.2 | 17.1±0.3 | 31.1±0.3 | - | - |
| ATT [22] | 48.3±1.0 | 67.7±0.6 | 74.5±0.9 | 26.1±0.5 | 44.2±0.8 | 51.2±0.2 | 11.0±0.4 | 25.8±0.7 | 4.7±1.4 | 8.7±1.0 |
| MCT [54] | 48.5±0.2 | 66.0±0.3 | 72.3±0.3 | 24.5±0.5 | 42.5±0.5 | 46.8±0.2 | 9.6±0.5 | 22.6±0.8 | - | - |
| NCFM [44] | 49.5±0.3 | 71.8±0.3 | 77.4±0.3 | 34.4±0.5 | 48.7±0.3 | 54.7±0.2 | 18.2±0.5 | 26.8±0.6 | - | - |
| *Inner-loop Optimization* | | | | | | | | | | |
| BPTT [46] | 49.1±0.6 | 62.4±0.4 | 70.5±0.4 | 21.3±0.6 | 34.7±0.5 | - | - | - | 1.1±0.7 | 2.3±0.9 |
| FRePO [55] | 45.6±0.1 | 63.5±0.1 | 70.7±0.1 | 26.3±0.1 | 41.3±0.1 | 41.5±0.1 | 16.9±0.1 | 22.4±0.1 | - | - |
| RCIG [24] | 49.6±1.2 | 66.8±0.3 | - | 35.5±0.7 | - | - | 22.4±0.3 | - | - | - |
| RaT-BPTT [9] | 53.2±0.7 | 69.4±0.4 | 75.3±0.3 | 35.3±0.4 | 47.5±0.2 | 50.6±0.2 | 20.1±0.3 | 24.4±0.2 | 5.2±1.1 | 13.0±0.9 |
| Teddy [49] | 30.1±1.4 | 53.0±0.5 | 66.1±0.4 | 13.5±0.4 | 33.4±0.7 | 49.4±0.5 | - | - | - | **34.1±0.8** |
| Ours | **54.4±0.6** | **72.4±0.3** | **78.7±0.2** | **36.9±0.5** | **49.0±0.6** | **55.9±0.1** | **24.3±0.4** | **32.7±0.5** | **14.7±0.7** | 30.6±0.3 |
| Full dataset | 84.8 | | | 56.2 | | | 37.6 | | 33.8 | |

## 5.3 Computational Efficiency Comparison

To evaluate the computational efficiency of our approach, we compare the average GPU memory consumption and training time among three leading data distillation methods: DATM [12], RaT-BPTT [9], and our proposed AT-BPTT. All experiments are conducted on NVIDIA A800 GPUs with IPC setting of 1, 10, and 50. As demonstrated in Fig. 4, AT-BPTT achieves superior computational efficiency while maintaining competitive performance, delivering a 63% reduction in memory usage and a $3.9\times$ speedup compared to RaT-BPTT. These results underscore the exceptional scalability and practical viability of our method for real-world deployment.

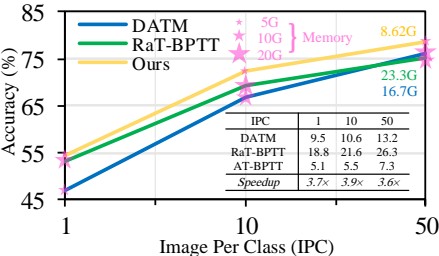

Figure 4: Comparison of performance, GPU memory usage, and speedup between the SOTA DD methods and our AT-BPTT.

## 5.4 Ablation Study

**Component Combination Evaluation.** In this section, we evaluate the individual and combined contributions of three performance components: dynamic truncation position (DTP), adaptive window size (AWS), and patch-wise semantic preservation (PSP). Ablation studies are conducted under the IPC=10 setting across CIFAR-10/100 [15], Tiny-ImageNet [16] and ImageNet-1K [29]. Tab. 2 shows that the integrated application of DTP and AWS yields a combined optimization effect, resulting in a 2.8% accuracy improvement, which surpasses the sum of the individual contributions. This arises from the complementary information processing mechanisms of DTP and AWS, which enhance the model's ability to capture key features of the original dataset. Regarding the PSP component, Tab. 2 reveals that its accuracy improvement is marginal on CIFAR-10/100 yet it is substantial on two datasets with higher resolution. When combined with DTP and AWS, it leads to remarkable gains of 8.3% and 17.6%. This is attributed to PSP's ability to segment high-resolution images into smaller patches, thereby fully leveraging AT-BPTT's strengths across various resolutions.

**Stage Transition Threshold Analysis.** To achieve precise control over stage transitions during model training, we study four critical hyperparameters: $M$ (the accuracy variation threshold in the early stage), $X$ (the counter threshold for early stage), $N$ (the accuracy variation threshold in the middle stage), and $Y$ (the counter threshold for middle stage). Through experiments with the CIFAR-10 [15]

Table 2: Ablation study for the contribution of different components in our framework. The improvements denoted by red numbers are with respect to baseline.

| DTP | AWS | PSP | CIFAR-10 | CIFAR-100 | Tiny-ImageNet | ImageNet-1K |
|---|---|---|---|---|---|---|
| × | × | × | 69.4 | 47.5 | 24.4 | 13.0 |
| ✓ | × | × | 70.7 (↑ 1.3) | 48.1 (↑ 0.6) | 27.3 (↑ 2.9) | 19.7 (↑ 6.7) |
| × | ✓ | × | 70.2 (↑ 0.8) | 47.9 (↑ 0.4) | 26.6 (↑ 2.2) | 17.9 (↑ 4.9) |
| × | × | ✓ | 69.5 (↑ 0.1) | 47.8 (↑ 0.3) | 25.4 (↑ 1.0) | 17.2 (↑ 4.2) |
| ✓ | ✓ | × | 72.2 (↑ 2.8) | 48.7 (↑ 1.2) | 30.1 (↑ 6.7) | 23.4 (↑ 13.4) |
| ✓ | ✓ | ✓ | **72.4** (↑ 3.0) | **49.0** (↑ 1.5) | **32.7** (↑ 8.3) | **30.6** (↑ 17.6) |

Table 3: Ablation study for $d$ corresponding to window size. The improvements denoted by red numbers are with respect to baseline.

| $d$ | Accuracy (%) | Time (hours) | Memory (GB) |
|---|---|---|---|
| 0 | 70.56 | 4.9 | 5.34 |
| 5 | 71.43 (↑ 0.87) | 5.1 (↑ 0.2) | 5.62 (↑ 0.28) |
| 10 | 72.44 (↑ 1.88) | 5.5 (↑ 0.6) | 6.14 (↑ 0.80) |
| 15 | 72.76 (↑ 2.2) | 5.8 (↑ 0.9) | 7.38 (↑ 2.04) |
| 20 | 72.95 (↑ 2.39) | 6.2 (↑ 1.3) | 9.16 (↑ 3.82) |

Figure 5: Ablation study for the stage transition threshold. The left and right matrices reflect the effect of $X$ and $M$, and $Y$ and $N$ on the accuracy, respectively. Darker colored squares indicate higher accuracy under the synergistic influence of horizontal and vertical coordinates.

under IPC=10, we identify that suboptimal parameter configurations induce two failure modes: the plateaus in the initial training phase or too early transition to the final stage, both of which disrupt the staged paradigm and cause substantial performance degradation. Fig. 5 demonstrates optimal parameterization when $X$ corresponds to 5% of total training epochs and $Y$ to 4%, with $M$=1.5 and $N$=1.0. This configuration achieves balanced stage duration distribution, stable convergence patterns, and better AT-BPTT performance.

**Window Size Analysis.** The hyperparameter $d$ in Section 4.2 governs the adjustment range of the truncation window, demonstrating a trade-off between model performance and computational efficiency. We conduct an ablation study on CIFAR-10 [15] with IPC=10 and quantitatively assess the dual effects of varying $d$ values. As Tab.3 shows, progressive increases in $d$ simultaneously enhance test accuracy and computational demands. Notably, compared to the baseline ($d = 0$), $d = 10$ achieves a 1.88% accuracy improvement with modest resource increments, with only 0.6 hours additional training time and 0.8 GB GPU memory usage. However, when $d > 10$, the accuracy gains diminish while the computational cost rises substantially. This mainly stems from the need for longer timesteps and the corresponding exponential growth in the dimensionality of the Hessian matrix as $d$ increases. Based on a trade-off analysis between performance gains and computational costs, we set the value of $d$ to 10 and report the results obtained with this setting. More ablation studies are provided in Appendix C.

## 6 Conclusion

This paper proposes AT-BPTT, a novel dataset distillation framework that optimizes inner-loop training through staged adaptation. Our analysis reveals that neural networks prioritize distinct learning patterns across training stages, necessitating adaptive truncation mechanisms. AT-BPTT resolves this by dynamically aligning truncation positions with gradient magnitude distributions, adjusting window sizes based on gradient stability, and reducing computational cost by low-rank Hessian approximation. The experiments across four benchmark datasets confirm that our method significantly outperforms existing dataset distillation method, achieving superior accuracy with efficient training. Furthermore, the formalization of gradient dynamics as a stage indicator provides a theoretical foundation for future research in bilevel optimization. Future work will focus on extending AT-BPTT to recurrent architectures and federated learning scenarios.

## Acknowledgments

This work is supported in part by the Natural Science Foundation of Sichuan Province (Grant No. 2025ZNSFSC1464), the China Postdoctoral Science Foundation (Grant No. 2024M760357), the Postdoctoral Fellowship Program of CPSF (Grant No. GZB20240115), the Sichuan Science and Technology Program (granted No. 2024ZDZX0011), the Fundamental Research Funds for the Central Universities No.ZYGX2025XJ042 and the Natural Science Foundation of China (Grant No. 62406057).

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

**Algorithm 1:** Automatic Truncated Backpropagation Through Time
---
**Input:** Training dataset $\mathcal{D}$, model parameters $\theta$, total training timesteps $T$.
**Params:** Accuracy variation thresholds $M, N$, transition counters $X, Y$, temperature
parameter $\tau$.

**1** Initialize zero-initialized counters $C_1, C_2$.
**2 for** $t = 1, 2, ..., T$ **do**
**3**     Segment images through patch-wise semantic preservation.
**4**     Compute accuracy variation $\Delta A_t = A_t - A_{t-1}$.
**5**     Update counters: $C_1 \leftarrow C_1 + \mathbf{1}(\Delta A_t < M)$, $C_2 \leftarrow C_2 + \mathbf{1}(\Delta A_t < N)$.
**6**     **if** $C_1 \geq X$ **then**
**7**       Switch to **Middle Stage**.
**8**     **if** $C_2 \geq Y$ **then**
**9**       Switch to **Late Stage**.
**10**     Compute gradient magnitude $\|\nabla_\theta \mathcal{L}_t\|_2$.
**11**     Computes probability $P_{\text{trunc}}(t)$.
**12**     Adjust window size $W^*(t)$ computed with $d$.
**13**     Compute $G_{\text{AT-BPTT}}$ through low-rank Hessian approximation.
**14**     Compute inner-loop loss $\mathcal{L}(\theta_T(\mathcal{S}))$.
**15**     Update model: $\theta_{t+1} \leftarrow \theta_t - \alpha \nabla_\theta \mathcal{L}_t$.
**16**     Compute outer-loop loss $\mathcal{L}_{\text{meta}}$ on validation set.
**17**     Compute meta-gradient $\nabla_\mathcal{S} \mathcal{L}_{\text{meta}}$.
**18**     Update distilled dataset: $\mathcal{S} \leftarrow \mathcal{S} - \alpha' \nabla_\mathcal{S} \mathcal{L}_{\text{meta}}$.
**Output:** Distilled dataset $\mathcal{S}$.
---

## A The Mathematical Derivation for Equation.2 in Main Text

We minimize the test loss $\mathcal{L}(\theta_T(\mathcal{S}), \mathcal{D})$, where $\theta_T(\mathcal{S})$ is the model parameter obtained through inner optimization on the synthetic dataset $\mathcal{S}$. Update parameters via gradient descent:

$$\theta_{t+1} = \theta_t - \alpha \nabla_\theta \mathcal{L}(\theta_t, \mathcal{S}), \tag{17}$$

for $T$ steps, resulting in $\theta_T(S)$. The goal of BPTT is to compute the gradient of the outer loss with respect to $\mathcal{S}$, i.e., the meta-gradient $\frac{\partial \mathcal{L}}{\partial \mathcal{S}}$. We express $\theta_T$ as the result of $T$ updates from the initial parameters $\theta_0$ :

$$\theta_T = \theta_0 - \alpha \sum_{t=0}^{T-1} \nabla_\theta \mathcal{L}(\theta_t(\mathcal{S}), \mathcal{S}). \tag{18}$$

The meta-gradient is decomposed into:

$$\frac{\partial \mathcal{L}}{\partial \mathcal{S}} = \underbrace{\frac{\partial \mathcal{L}}{\partial \theta_T}}_{\text{Outer gradient}} \cdot \underbrace{\frac{\partial \theta_T}{\partial \mathcal{S}}}_{\text{Inner gradient propagation}} . \tag{19}$$

We then expand the parameter update process recursively:

$$\frac{\partial \theta_{t+1}}{\partial \mathcal{S}} = \frac{\partial \theta_t}{\partial \mathcal{S}} - \alpha \left[ \frac{\partial}{\partial \mathcal{S}} \nabla_\theta \mathcal{L}(\theta_t, \mathcal{S}) \right] = \frac{\partial \theta_t}{\partial \mathcal{S}} - \alpha \left[ \underbrace{\nabla_\theta^2 \mathcal{L}(\theta_t, \mathcal{S}) \cdot \frac{\partial \theta_t}{\partial \mathcal{S}}}_{\text{Hessian term}} + \underbrace{\nabla_\theta \nabla_\mathcal{S} \mathcal{L}(\theta_t, \mathcal{S})}_{\text{Mixed derivative term}} \right]. \tag{20}$$

We expand the recursion into an explicit summation:

$$\frac{\partial \theta_T}{\partial \mathcal{S}} = -\alpha \sum_{t=0}^{T-1} \prod_{j=t+1}^{T-1} \left[ 1 - \alpha \frac{\partial^2 \mathcal{L}(\theta_j(\mathcal{S}), \mathcal{S})}{\partial \theta^2} \right] \cdot \frac{\partial^2 \mathcal{L}(\theta_t(\mathcal{S}), \mathcal{S})}{\partial \theta \partial \mathcal{S}}. \tag{21}$$

The final BPTT Meta-Gradient is formulated as:

$$\mathcal{G}_{BPTT} = -\alpha \frac{\partial \mathcal{L}(\theta_T(\mathcal{S}), \mathcal{D})}{\partial \theta_T} \sum_{t=0}^{T-1} \prod_{j=t+1}^{T-1} \left[ 1 - \alpha \frac{\partial^2 \mathcal{L}(\theta_j(\mathcal{S}), \mathcal{S})}{\partial \theta^2} \right] \cdot \frac{\partial^2 \mathcal{L}(\theta_t(\mathcal{S}), \mathcal{S})}{\partial \theta \partial \mathcal{S}}. \tag{22}$$

# B    More Implementation details

To validate the effectiveness of our method while ensuring experimental rigor, we employ a controlled variable approach in designing the experimental protocol. The critical experimental parameters and configurations are rigorously aligned with those of RaT-BPTT [9], specifically encompassing essential aspects including data preprocessing procedures, model convergence acceleration mechanisms, gradient explosion mitigation strategies, and label processing methodologies. Each reported result is the average of 5 experiments..

**Data Preprocessing.** In the experimental setup, all datasets were subjected to a unified geometric augmentation strategy comprising random rotation and horizontal flipping operations. The augmented data subsequently underwent ZCA whitening processing with a regularization coefficient $\lambda = 0.1$. For synthetic data preprocessing, the initial data generation was performed through a Gaussian distribution-based random initialization method, followed by normalization operations to ensure numerical distribution consistency.

**Model Convergence Acceleration Mechanisms.** During the distillation stage, three acceleration strategies are implemented to enhance optimization efficiency: (1) An Exponential Moving Average (EMA) technique is adopted to accelerate model convergence while improving stability and generalization capabilities; (2) The Higher framework is employed for efficient meta-gradient computation; (3) An Adam optimizer with learning rate $\alpha = 0.001$ is utilized for precise inner-loop updates.

**Gradient Explosion Mitigation and Label Processing.** To address the issue of gradient explosion, our methodology incorporates a meta gradient clipping strategy while ensuring gradient stability through sufficiently large batch sizes for each dataset. Consistent with the experimental configuration of RaT-BPTT, our approach maintains architectural coherence by employing raw positive real-value representations rather than implementing normalization on label probability distributions.

**Hyperparameter Settings.** In the experiments, several hyperparameters require tuning, as they directly influence the distillation performance of the method. Accordingly, we detail the hyperparameter settings used for performance evaluation in the main text as presented in the Tab. 4. Specifically, window denotes the initial window size, totwindow represents the total number of unrolled time steps, $d$ controls the range of the truncation window size, $lr$ indicates the learning rate, architecture refers to the adopted network architecture, batch size determines the number of training samples used in a single parameter update, and epoch specifies the number of iterations.

Table 4: Description of Hyperparameters in Experiments.

| Dataset | CIFAR-10 | CIFAR-100 | Tiny-ImageNet | ImageNet-1K |
|---|---|---|---|---|
| window | 40 | 60 | 100 | 120 |
| totwindow | 200 | 200 | 300 | 400 |
| $d$ | 10 | 20 | 40 | 50 |
| $lr$ | 0.001 | 0.001 | 0.0003 | 0.0001 |
| architecture | ConvNet3 | ConvNet3 | ConvNet4 | ConvNet5 |
| batch size | 1000 | 1000 | 500 | 250 |
| epoch | 600 | 600 | 1000 | 1000 |

# C    More Experimental Results

## C.1    Hypothesis Verification on Tiny-ImageNet

We conduct similar verification experiments on Tiny-ImageNet [16]. As shown in Fig. 6, the Tiny-ImageNet exhibits three-stage characteristics consistent with CIFAR-10: (1) During the early stage, models preferencely select preliminary truncation positions with larger window sizes; (2) middle stage demonstrates limited performance sensitivity to variations in both truncation positions and window sizes; (3) In the late stage, models shift preference towards post truncation positions while window size adjustments show diminishing impact. This cross-dataset consistency substantiates the generalizability of our findings and confirms the effectiveness of the AT-BPTT across diverse datasets.

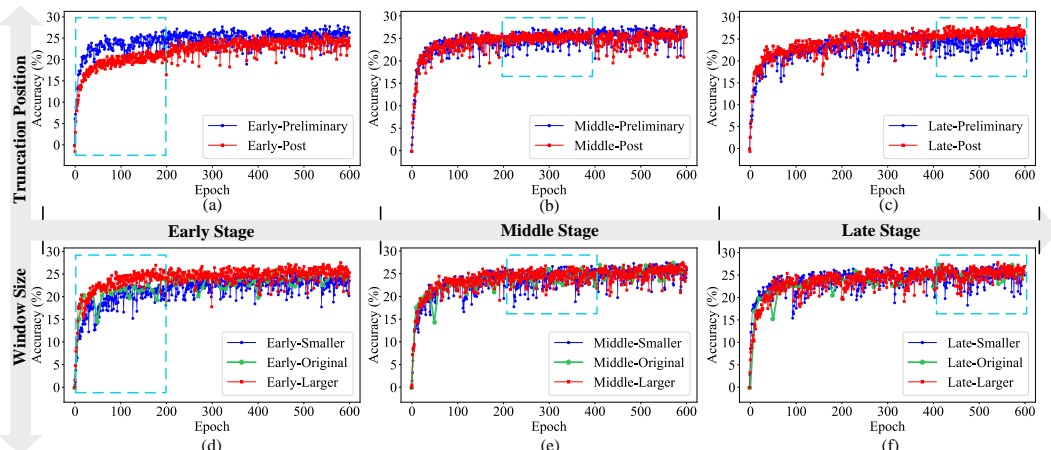

Figure 6: Hypothesis verification for the truncation strategies and window size on Tiny-ImageNet [16]. (a)(b)(c) show experiments where the preliminary or post truncation positions are implemented at early, middle and late stages, respectively, and (d)(e)(f) present experiments where the window size is changed after fixing the truncation position. For example, Early-Preliminary in (a) means that randomly select preliminary phase (0-100) timesteps in early training stage (0-200 epochs).

Table 5: Computational efficiency and performance comparison with low-rank Hessian approximation (LRHA) on CIFAR-10 when IPC=10.

| Method | GPU Memory (GB) | Training Time (h) | Accuracy (%) | Memory Saving | Speedup |
|---|---|---|---|---|---|
| RaT-BPTT [9] | 18.9 | 21.6 | 69.4 | – | 1.0× |
| AT-BPTT (w/o LRHA) | 15.3 | 18.4 | **72.7** | 19.0% | 1.17× |
| AT-BPTT (w/ LRHA) | **6.94** | **5.5** | 72.4 | **63.3%** | **3.9×** |

## C.2 Impact of Low-Rank Hessian Approximation

To rigorously evaluate the impact of our low-rank Hessian approximation (LRHA) mechanism, we conduct controlled experiments comparing three configurations on CIFAR-10 with IPC=10: (1) Baseline RaT-BPTT with full Hessian computation, (2) AT-BPTT without LRHA, and (3) Full AT-BPTT with LRHA enabled. All experiments are performed on NVIDIA A800 GPUs with identical hyperparameters and ConvNet-3 architecture. The results in Tab. 5 demonstrate that LRHA provides substantial computational benefits while preserving model performance.

## C.3 Generalization Experiments on Wide Networks

Previous studies [55, 24] have sought to narrow the discrepancy between proxy training environments and real training scenarios by adopting wider network architectures. Notably, the RaT-BPTT [9] method similarly employs a paradigm of training on narrow networks while evaluating on wide networks (4 times wider ConvNet). This study systematically evaluates the performance of AT-BPTT under wide network architectures through experiments. Experimental results as shown in Tab.6 demonstrate that AT-BPTT maintains its comprehensive leading advantage over RaT-BPTT in wide network configurations, outperforming RaT-BPTT by an average of 3.4%. Compared with narrow network configurations, AT-BPTT achieves measurable performance improvements in wide architectures. This outcome successfully validates prior hypotheses and directly stems from AT-BPTT's phased training strategy with dynamically adjusted windows, confirming the effectiveness of its adaptive mechanism.

## C.4 Comparison with Diffusion Model-Based Methods

In addition to optimization methods based on the inner- and outer-loop, generative model-based methods have been developed for synthesizing compact datasets. These methods leverage Generative Adversarial Networks (GANs) or diffusion models to produce high-quality synthetic data, gaining

Table 6: Comparison with the baseline dataset distillation methods on wide networks. The improvements denoted by red numbers are with respect to our baseline RaT-BPTT*. Each reported result is the average of 5 experiments.

| Dataset | CIFAR-10 | | | CIFAR-100 | | | Tiny-ImageNet | | AVG |
|---|---|---|---|---|---|---|---|---|---|
| Img/class(IPC) | 1 | 10 | 50 | 1 | 10 | 50 | 1 | 10 | |
| KIP* [28] | 49.9±0.2 | 62.7±0.3 | 68.6±0.2 | 15.7±0.2 | 28.3±0.1 | - | - | - | - |
| RFAD* [23] | 53.6±1.2 | 66.3±0.5 | 71.1±0.4 | 26.3±1.1 | 33.0±0.3 | - | - | - | - |
| FRePO* [55] | 46.8±0.7 | 65.5±0.6 | 71.7±0.2 | 28.7±0.1 | 42.5±0.2 | 44.3±0.2 | 15.4±0.3 | 25.4±0.2 | 42.5 |
| RCIG* [24] | 53.9±1.0 | 69.1±0.4 | 73.5±0.3 | 39.3±0.4 | 44.1±0.4 | 46.7±0.1 | 25.6±0.5 | 29.4±0.9 | 47.7 |
| RaT-BPTT* [9] | 54.1±0.4 | 71.0±0.2 | 75.4±0.2 | 36.5±1.0 | 47.9±0.2 | 51.0±0.6 | 20.3±0.9 | 24.9±1.3 | 47.6 |
| Ours* | **55.6±0.3** | **71.8±0.2** | **78.7±0.2** | **37.2±0.5** | **49.9±0.7** | **54.7±0.2** | **26.8±0.8** | **33.4±0.6** | **51.0** |
| Δ% | +1.5 | +0.8 | +3.3 | +0.7 | +2.0 | +3.7 | +6.5 | +8.5 | +3.4 |

significant popularity and advancements in recent years. We select two leading diffusion model-based methods, $D^4M$ [34] and $D^2M$ [30], as baselines and evaluate the distillation performance of AT-BPTT under identical experimental settings. As shown in Tab. 7, AT-BPTT maintains a leading position in distillation performance, achieving a average of 6.7% performance improvement across various datasets.

Table 7: Comparison with decoupled optimization-based distillation methods using CIFAR-10/100 and Tiny-ImageNet datasets, with bolded values indicating the highest test accuracy.

| Dataset | CIFAR-10 | | | CIFAR-100 | | | Tiny-ImageNet | | |
|---|---|---|---|---|---|---|---|---|---|
| Img/class(IPC) | 1 | 10 | 50 | 1 | 10 | 50 | 1 | 10 | 50 |
| $D^4M$ [34] | - | 56.2 | 72.8 | - | 45.0 | 48.8 | - | - | 46.8 |
| $D^2M$ [30] | 50.2 | 67.8 | 74.4 | 29.8 | 46.6 | 51.2 | 16.7 | 26.1 | 30.1 |
| Ours | **54.4** | **72.4** | **78.7** | **36.9** | **49.0** | **55.9** | **24.3** | **32.7** | **48.7** |

## C.5 Comparison with Baseline on Other Datasets

To further demonstrate the effectiveness of AT-BPTT on high-resolution dataset, we conduct experiments on ImageNet subsets [13], including ImageNette, ImageWoof, ImageMeow, and ImageFruit. The experiments compare AT-BPTT with baseline methods, including Random MTT [2], RDED [36], FTD [8], DATM [12], EDF [40], and NCFM [44]. As presented in the Tab. 8, AT-BPTT consistently achieves the highest accuracy across all datasets and IPC settings, for example, attaining 79.1% on ImageNet (IPC=10) and 47.6% on ImageFruit (IPC=1). These results significantly outperform leading methods, demonstrating that AT-BPTT exhibits superior generalization and performance stability across diverse datasets and data scales. This advantage is particularly pronounced in data-constrained scenarios (IPC=1), underscoring its efficacy in image classification tasks.

Table 8: Comparison with baseline methods using ImageNette, ImageWoof, ImageMeow, and ImageFruit dataset [13], with bolded values indicating the highest test accuracy.

| Dataset | ImageNette | | ImageWoof | | ImageMeow | | ImageFruit | |
|---|---|---|---|---|---|---|---|---|
| Img/class(IPC) | 1 | 10 | 1 | 10 | 1 | 10 | 1 | 10 |
| Random [53] | 23.5 | 47.7 | 14.2 | 27.0 | 13.8 | 29.0 | 13.2 | 21.4 |
| MTT [2] | 47.7 | 63.0 | 28.6 | 35.8 | 30.7 | 40.4 | 26.6 | 40.3 |
| RDED [36] | 33.8 | 63.2 | 18.5 | 40.6 | - | - | - | - |
| FTD [8] | 52.2 | 67.7 | 30.1 | 38.8 | 33.8 | 43.3 | 29.1 | 44.9 |
| DATM [12] | 52.5 | 68.9 | 30.4 | 40.5 | 34.0 | 48.9 | 30.9 | 45.5 |
| EDF [40] | 52.6 | 71.0 | 30.8 | 41.8 | 34.5 | 52.6 | 32.8 | 46.2 |
| NCFM [44] | 53.4 | 77.6 | 27.2 | 48.4 | 34.6 | 58.2 | 29.2 | 44.8 |
| Ours | **55.5** | **79.1** | **31.8** | **49.8** | **36.0** | **60.8** | **34.1** | **47.6** |

## C.6 Experiments on Language Tasks

To assess AT-BPTT's generalization capability for language tasks, we evaluate our method on three standard text classification benchmarks: SST-2 [39], MNLI-m [39], and AGNews [50]. As

demonstrated in Table 9, AT-BPTT achieves consistent improvements over existing textual dataset distillation approaches, with an average accuracy gain of 3.2% across all evaluated datasets. The results demonstrate that our method is also effective on non-visual datasets.

Table 9: Comparison with the SOTA textual dataset distillation methods on BERT [7] across SST-2 [39], MNLI-m [39], and AGNews [50].

| Dataset | SST-2[39] | | | MNLI-m[39] | | | AGNews[50] | | |
|---------|------|------|------|------|------|------|------|------|------|
| Img/class(IPC) | 1 | 10 | 50 | 1 | 10 | 50 | 1 | 10 | 50 |
| DDAL [26] | 64.5 | 69.7 | 73.5 | 34.1 | 38.8 | 41.5 | 26.1 | 28.9 | 34.2 |
| DiLM [27] | 72.5 | 76.3 | 80.3 | 39.7 | 44.8 | 48.7 | 27.8 | 30.9 | 36.5 |
| DaLLME [37] | 72.3 | 77.5 | 80.9 | 42.7 | 47.2 | 51.4 | 27.6 | 30.8 | 36.6 |
| Ours | **73.2** | **79.4** | **82.4** | **44.9** | **48.1** | **53.2** | **28.8** | **31.1** | **39.3** |
| Full dataset | 92.7 | | | 86.7 | | | 94.6 | | |

## D  Difficulty Sample Conjecture

Our theoretical analysis reveals the learning characteristics of deep neural networks (DNN) [1]: during the initial training stage, the networks tend to rapidly capture easily identifiable simple patterns in data, while progressively shifting their focus to more complex and fine-grained feature representations as training advances. This discovery aligns theoretically with the data difficulty scoring mechanism proposed in PAD [21] based on the trajectories matching framework [2]. Specifically, the PAD innovatively introduces a difficulty scoring function that effectively quantifies the learning complexity of data samples for DNNs, thereby establishing a dynamic curriculum learning mechanism. Through intelligent scheduling strategies, this mechanism gradually introduces samples of increasing difficulty during expert trajectory training, demonstrating inherent consistency with the phased training paradigm of the AT-BPTT algorithm.

Building upon these theoretical connections, we propose the following hypothetical improvement: systematically ordering the original dataset through the difficulty scoring function and combining it with a phased progressive training strategy to construct a stepwise learning path from low-difficulty to high-difficulty samples. We suggest focusing on low-scoring samples during the initial training stage and progressively incorporating high-scoring complex samples as model capability improves. This structured knowledge progression mechanism is expected to establish a more refined model learning paradigm, thereby enhancing overall training effectiveness. The development of a more sophisticated and refined data preprocessing framework presents a promising direction for further investigation, which constitutes one of our key research priorities in future studies.

## E  Discussion

Although we evaluate AT-BPTT in the context of dataset distillation, the proposed algorithm is generally applicable to a broad range of bilevel optimization problems that involve unrolled gradient-based inner loops. In particular, AT-BPTT can serve as a drop-in replacement for standard or truncated BPTT in applications such as meta-learning, logical reasoning [25], tool learning [43], agent [42], differentiable neural architecture search (NAS), and personalized federated learning. These applications often rely on fixed-length unrolling or heuristic truncation strategies, which can be suboptimal or inefficient. In contrast, AT-BPTT adaptively adjusts both the truncation points and the weighting of intermediate gradients based on the optimization landscape, potentially leading to improved stability, reduced memory and compute costs, and better overall performance. We believe AT-BPTT offers a principled and generalizable framework for efficient bilevel optimization, and we leave its integration into these broader domains as promising directions for future work.

AT-BPTT also has some drawbacks that need to be addressed. First, this inner-loop optimization inevitably involves unfolding the learning trajectory during training. Although we have simplified the computation of the Hessian matrix product using a low-rank Hessian approximation method, the computational cost remains significant, leading to higher time complexity compared to NCFM [44]. Second, our base network is still limited to relatively simple models like CNNs, results in practical limitations. Therefore, our future work will focus on improving the computational efficiency of

the algorithm or combining the inner- and outer-loop optimization to reduce the unfolding of the learning trajectory, and exploring the implementation of our algorithm on larger-scale models, such as diffusion models and transformers.

# F    Visualizations of Distilled Dataset

In this section, we present visualizations of the distilled datasets obtained from various datasets. The IPC of the presented datasets is uniformly set to 10, with Fig. 7 illustrating the visualization results on CIFAR-10, respectively.

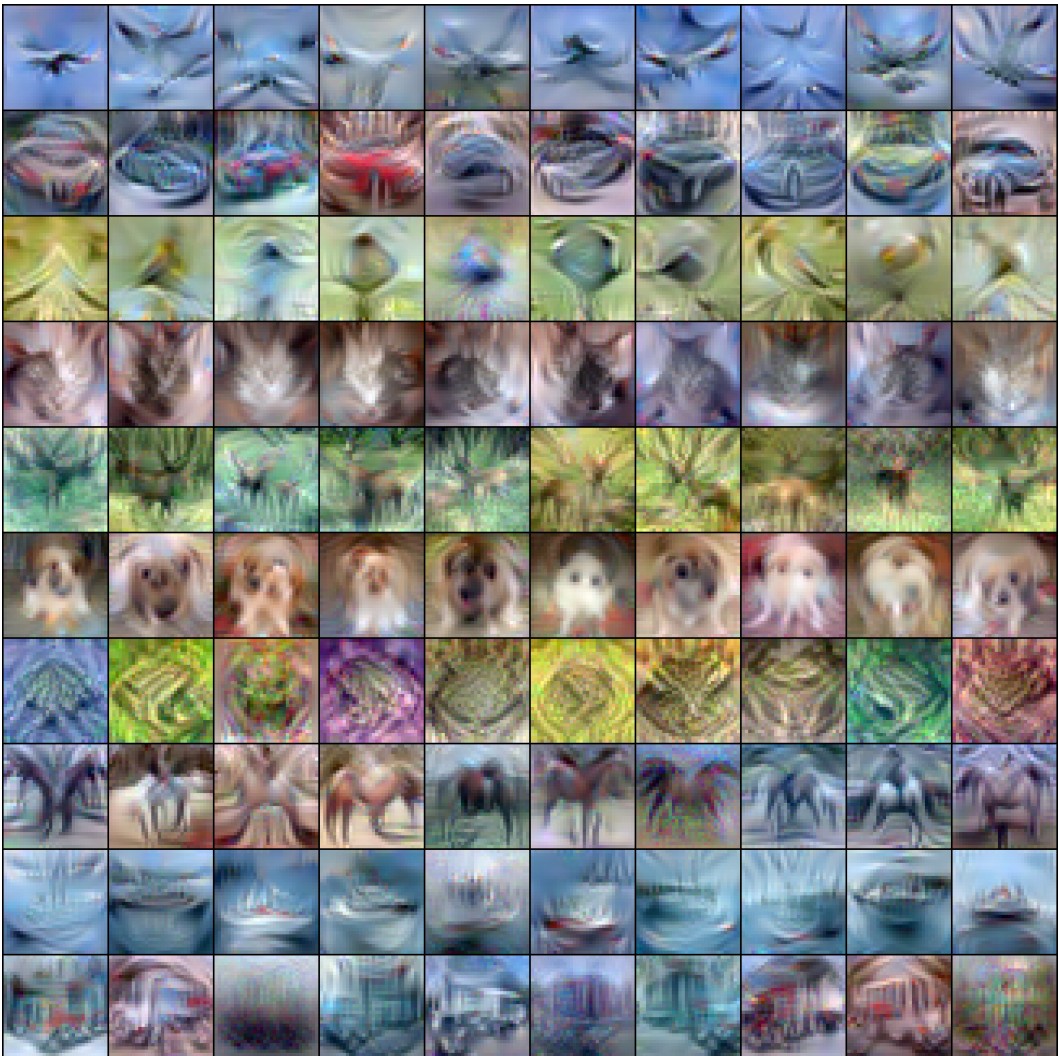

Figure 7: Visualization results of our proposed method for CIFAR-10 [15] with IPC=10.

# G    Ethical Statement

This research proposes a general optimization algorithm for bilevel problems that improves the efficiency and adaptability of truncated backpropagation through time. The work was conducted with integrity, transparency, and academic rigor. No human subjects, sensitive data, or personally identifiable information were involved. The method does not raise concerns regarding fairness, bias, or misuse in its current form. We have also ensured reproducibility by providing full methodological details and plan to release code under an open-source license upon publication.

## H  Broader Impacts

This paper introduces an adaptive and efficient method for truncated gradient backpropagation in bilevel optimization, which has the potential to significantly reduce computational costs in a range of machine learning applications, including meta-learning, federated learning, neural architecture search, and dataset distillation. The positive societal impact includes making advanced machine learning more accessible and sustainable by lowering the hardware and energy requirements of training large models. This aligns with ongoing efforts in the community to improve the environmental footprint of ML research.

