# OpenReview forum: "Beyond Random: Automatic Inner-loop Optimization in Dataset Distillation"
_NeurIPS.cc/2025/Conference — NeurIPS 2025 poster_

### Official Review · Reviewer_B86A · 2025-06-26

**Clarity:** 3
**Significance:** 3
**Originality:** 2
**Rating:** 5
**Confidence:** 3

**Summary:**

This paper focuses on dataset distillation. This technique aims to create a compact dataset that preserves model accuracy. Dataset distillation is formulated as a bilevel optimization problem where the inner loop is responsible for training on a small, synthetic dataset, and the outer loop minimizes the generalization error on the target dataset. This paper builds on RaT-BPTT which proposed to truncate and apply a window on inner loop gradient calculation.
The authors propose to dynamically set the truncation position based on gradient magnitude. They also introduce an adaptive window sizing strategy determined by gradient variation. Finally, they propose using a low-rank Hessian approximation to increase training speed and reduce memory cost. Extensive experiments on CIFAR-10, CIFAR-100, Tiny-ImageNet, and ImageNet-1K show substantial gains compared to baseline methods.

We can formulate 2 claims from the paper:
1. "Sampling specific timesteps from distinct stages lead to **better** performance, rather than uniformly applying random truncation" (n.b. optimal is misused in the paper)
2. "Adjusting the truncation window size further enhance distillation performance"

Then we can summarize the contributions as follows:
1. Introduce a dynamic truncation strategy and window sizing based on outer loop stages
2. Introduce a low-rank Hessian approximation to reduce memory and computational cost
3. Patch-wise semantic preservation to ensure semantic consistency in high-resolution images

**Questions:**

Question 1: For clarity, could the authors explicitly define the scope of the 'Early,' 'Middle,' and 'Late' stages versus the 'Preliminary' and 'Post' phases? My understanding, inferred from the text (e.g., Section 4.1 lines 122-124 ), is that 'Early,' 'Middle,' and 'Late' refer to distinct stages within the outer-loop training process (epochs), while 'Preliminary' and 'Post' describe sub-phases within the unrolled learning trajectory of each inner-loop epoch (timesteps). Usage of "inner-loop" and "outer-loop" terminology would really help clarify this distinction.

Then, if I'm not mistaken,

Question 2: To strengthen the reproducibility and practical applicability of AT-BPTT, could the authors provide a more detailed analysis or discussion on the hyperparameter tuning process? Specifically, given the numerous new hyperparameters introduced (temperature τ and T for dynamic truncation, temperature τ and d for adaptive window size, thresholds M,N,X,Y for stage transition, and kmax​,kmin​ for LRHA ), how were these values determined for CIFAR-100, Tiny-ImageNet, and ImageNet-1K, considering the ablation studies in Section 5.4 only focus on CIFAR-10?

It is stated that AT-BPTT is 'automatic', yet the extensive set of tunable parameters suggests a potentially high computational cost and time investment for hyperparameter search on new datasets.  We would appreciate a discussion or, ideally, an empirical evaluation of:

    The sensitivity of AT-BPTT's performance to these hyperparameters across different datasets.
    The computational overhead associated with tuning these hyperparameters for each dataset.
    Any strategies or heuristics employed to efficiently set these parameters for new datasets without extensive grid search, perhaps leveraging insights from the intrinsic gradient behaviors observed.

This would greatly clarify the true 'automatic' nature and practical deployment feasibility of the proposed framework.

Question 3: In the experiments, the method "Teddy" seems to be performing well. Although, the authors do not mention Teddy in the related work section. It's also not compared in the table 7 which seems to be pretty close to the experiments in the Teddy paper. Could you detail how Teddy relates to your work and conducts additional experiments to compare with this paper?

Question 4: In all the results, and espacially on Tiny Imagenet and Imagenet 1K in table 1 (but also in annexes), the authors should report the accuracies with and without patch-wise semantic preservation. They should also display the results of one of the methods (RaT-BPTT) with and without patch-wise semantic preservation. This would help to understand each individual contribution on high resolution datasets.

**Ethical Concerns:**

["NO or VERY MINOR ethics concerns only"]

**Final Justification:**

I have engaged a conversation on papers weaknesses and the authors have addressed my concerns.

**Limitations:**

Yes

**Quality:**

3

**Strengths And Weaknesses:**

Strengths:

The results are quite significant, with AT-BPTT outperforming RaT-BPTT and other baselines on CIFAR-10, CIFAR-100, Tiny-ImageNet, and ImageNet-1K.

Weaknesses:

Clarity could be improved as mentioned on the questions below. It's not clear to me what is the link between paper cited as [1] on learning dynamics and the proposed method. It seems to be more of an intuitive connection rather than a formal one.

Dependence on hyperparameters is not well discussed. The paper claims that AT-BPTT is 'automatic', yet the extensive set of tunable parameters suggests a potentially high computational cost and time investment for hyperparameter search on new datasets.

Patch-Wise semantic preservation is a bit "on the side" of the main contribution, and it is not clear how it relates to the main claims of the paper. The issue is that it's hard to differentiate between the contribution of the patch-wise semantic preservation and the dynamic truncation and adaptive window sizing. The paper does not provide a clear separation of these contributions, making it difficult to assess their individual impacts on performance.

Formal issues:
Hypotheses given (1 and 2) are in fact research questions.

---

> ### Author Rebuttal · Authors · 2025-07-30
>
> Thank you for recognizing the significant performance gains of AT‑BPTT. Below we will address specific questions.
>
> > ***W1/Q1: Clarity and concerns about the link***
>
> Thank you for pointing out the need to clarify the definitions and the link between [1] and our method.
>
> We appreciate that your interpretation is consistent with our formal definitions of stages and phases. For clarity, we will provide the descriptions below to explicitly define three stages and two phases, and these will be provided in the main text:
> - **Outer-loop Training Stages:** We use **Early**, **Middle**, and **Late** to describe the global stages of the entire dataset distillation process. These stages are counted by epochs.
> - **Inner-loop Learning Phases:** We use '**Preliminary**' and '**Post**' to describe sub-phases within the unrolled learning trajectory. These phases are counted by timesteps.
>
> Additionally, we are sure that there is a profound inherent link between the paper [1] and our method. The essence of our three-stage division lies in focusing on the learning priorities of the model at different stages. This aligns exactly with the conclusion in [1]: early stages prioritize simple patterns and later stages refine complex features. Based on Reviewer aCEq's Q1, we can confidently state that our method not only has a strong formal link with paper [1], but also extends its theoretical framework to dataset distillation.
>
> > ***W2/Q2: Hyperparameter fine-tuning***
>
> Thank you and Review zMmc for your suggestion regarding hyperparameter fine-tuning. Based on your comments and Reviewer zMmc's W2/W3, we offer the following comprehensive reply.
>
> We agree that clarifying the hyperparameter sensitivity, computational overhead, and practical strategies for setting these parameters is crucial for reproducibility and real-world deployment. Therefore, we provide clear guidelines to minimize tuning effort. Our analysis divides these hyperparameters into three categories based on dataset sensitivity, and we discuss empirical basis, practical strategy and computational cost.
>
>
> ### Sensitivity Category
> - Low sensitivity: Fixed defaults suffice across datasets due to universal gradient dynamics
> - Medium sensitivity: Adjusted via simple scaling rules, minimizing need for per-dataset tuning
> - High sensitivity: Only M and N require dataset-specific tuning
>
> ### Empirical Basis and Practical Strategy
> |Hyperparameter|Sensitivity|Empirical Basis|Practical Strategy|
> |:-:|:-:|:-:|:-:|
> |$\tau$ (temperature for dynamic truncation and adaptive window)|Low Sensitivity|Gradient magnitude distributions are similar across dataset, which makes $\tau$ less sensitive to dataset. We conduct ablation study from CIFAR-10 to ImageNet-1K with $\tau$ = 1.0 without re-tuning. Accuracy varied <±0.5% when $\tau$ is perturbed ±0.5 on CIFAR-10/ImageNet-1K.|$\tau$ = 1.0 for all datasets.|
> |k_min (LRHA min rank)|Low Sensitivity|k_min is invariant to dataset. k_min = 1 worked for all tested datasets. Higher k_min will add unnecessary cost.|k_min = 1 for all datasets.|
> |k_max scale (LRHA max rank factor)|Low Sensitivity|Fixed scaling factor.|Not a tunable hyperparameter.|
> |$d$ (window adjustment range)|Medium Sensitivity|Window size exhibits a direct correlation with the total number of timesteps $T$. $T$ is related to dataset scale. Larger datasets require larger windows to capture complex dynamics. |$d \in [10,50]$ for the scale of CIFAR-10 to ImageNet-1K|
> |X, Y (transition counters)|Medium Sensitivity|X and Y depend on training duration. We thus set X, Y proportionally to total epochs.|X = 15% of total epochs; Y = 12.5% of total epochs.|
> |M, N (accuracy variation thresholds)|High Sensitivity|M and N depend on convergence speed of dataset. Different datasets require different thresholds: e.g., M=1.5 for CIFAR-10, but M=1.0 for Tiny-ImageNet due to slower early convergence.|Ablation study in **Stage Transition Threshold Analysis** (lines 295-304).|
>
> ### Computational Cost of Hyperparameter Tuning
>
> The computational cost for tuning AT-BPTT hyperparameters is low, as most parameters are low-sensitivity or medium-sensitivity. As shown in table above, the costs of low-sensitivity or medium-sensitivity hyperparameters are negligible. The hyperparameter tuning is thus focused on M and N. And this process for other datasets is similar to that of ablation study in **Figure 5**. The computational costs on Nvidia A800 are shown in table below:
>
> | Dataset |CIFAR-10  | CIFAR-100| Tiny-ImageNet | ImageNet-1K|
> |:-:|:-:|:-:|:-:|:-:|
> |Time (h) | <25 | <27 | <40 | <80 |
> |Memory (GB) | <7 | <9 | < 18 | < 44 |
>
> If the paper is accepted, we will provide the analysis above in the camera-ready version.
>
> >***W3/Q4: Ablation study of PSP (or LRHA?)***
>
> Thank you for your comments on the Patch-wise Semantic Preservation (PSP) module. Our intention is not to design PSP as an inseparable component of AT-BPTT, but rather as a plug-and-play paradigm solving the challenges when applied to high-resolution data. To clearly differentiate the individual impact of three components, we have already conducted a detailed ablation study (lines 283-294), with the results presented in **Table 2**. In fact, the results of Tiny-ImageNet and ImageNet-1K are included in **Table 2**. Additionally, the results of RaT-BPTT with and without PSP are also included in the second and fifth lines in **Table 2**, since RaT-BPTT is our backbone.
>
> However, based on the questions you've raised, particularly "the authors should report the accuracies with and without patch-wise semantic preservation", we infer that there might be a typo in your comments. Could you please clarify whether you are referring to the Low-rank Hessian approximation (LRHA) instead of PSP in W3/Q4?
>
> To fully address your concerns, we have further conduct relevant experiments for LRHA that might meet your requirements. Since we have conducted its ablation study for low-resolution datasets in **Appendix C.2**, the results shown below are the extention of **Table 5**.
>
> | Dataset       | IPC | Method            | GPU Memory (GB) | Training Time (h) | Accuracy (%) |
> |:-:|:-:|:-:|:-:|:-:|:-:|
> |Tiny-ImageNet |   10  | RaT-BPTT    | 43.4    | 37.6    | 24.2   |
> |     |     | RaT-BPTT(w/ LRHA)   | 24.7    | 13.6     | 23.3    |
> | |   | AT-BPTT(w/o LRHA) | 38.7    | 28.5    | 33.2   |
> |     |     | AT-BPTT(w/ LRHA)  | 17.6   | 7.8    | 32.7  |
> | ImageNet-1K  |  10   | RaT-BPTT  | 85.4   | 71.4      | 12.8  |
> |    |     | RaT-BPTT(w/ LRHA)          | 51.4       | 22.5       | 11.2   |
> |    |   | AT-BPTT(w/o LRHA) | 72.8    | 62.4       | 31.4         |
> |      |     | AT-BPTT(w/ LRHA)  | 43.3    | 15.6      | 30.6         |
>
> Table above demonstrates that LRHA can significantly reduce memory and training time while only slightly reducing accuracy. If the paper is accepted, we will provide the ablation experiments above in the camera-ready version.
>
> >***Q3: Relationship and comparison with Teddy***
>
> Thank you for your suggestion to add details about Teddy [2] method. Teddy is an interesting DD method, and both it and our method have their own strengths. We will elaborate on the relationship and comparison between Teddy and our work.
>
> ### Relationship:
> Both methods have made improvements to reduce computational resource consumption in the meta-learning optimization objective. For the second-order derivative in the meta-learning objective, Teddy converts it to a first-order derivative using a Taylor approximation, while AT-BPTT approximates the Hessian matrix generated in the second-order derivative using the LRHA module.
> The methodologies of the two are complementary. Teddy uses Taylor approximation and model pooling to optimize global distillation, making it suitable for distillation with large-scale data; while AT-BPTT uses dynamic truncation to focus on fine-grained control of training dynamics, making it suitable for various inner-loop optimization scenarios.
>
> ### Comparison:
> Since Teddy is not open-sourced, we use the results of our reproduction for Teddy. We compare Teddy with our AT-BPTT in three aspects: test accuracy, cross-architecture, and high-IPC settings.
>
> - **Test accuracy:** As shown in **Table 1**, although our method is lower than Teddy on high-resolution datasets, it significantly outperforms Teddy on various low-resolution datasets.
>
> - **Cross-architecture:** Under similar settings in **Appendix C.4**, we conduct cross-architecture evaluations for Teddy and ours. Table below shows that AT-BPTT consistently outperforms Teddy across four tested architectures.
>
> | Architecture | ConvNet |      | VGG11 |      | AlexNet |      | ResNet18 |      |
> |:-:|:-:|:-:|:-:|:-:|:-:|:-:|:-:|:-:|
> | *IPC*          | 10      | 50   | 10    | 50   | 10      | 50   | 10       | 50   |
> | Teddy        | 53.0    | 66.1 | 46.2  | 57.7 | 39.3    | 55.4 | 51.8     | 63.4 |
> | Ours         | 72.3    | 78.9 | 68.9  | 77.1 | 68.5    | 76.3 | 69.3     | 75.6 |
>
> - **High-IPC settings:** Under similar settings in **Appendix C.5**, we conduct evaluations in high-IPC settings for Teddy and ours. Table below shows that AT-BPTT consistently outperforms Teddy across various high-IPC cases.
>
> | Dataset |   | CIFAR-10 |  |  |  | CIFAR-100 |  |  | Tiny-ImageNet |  |
> |:-:|:-:|:-:|:-:|:-:|:-:|:-:|:-:|:-:|:-:|:-:|
> |   *IPC*   |  250 |    500   | 1000 | 1500 |  25  |     50    |  100 |  150 |       50      |  100 |
> |  Teddy  | 81.7 |   84.2   | 86.5 | 90.4 | 51.5 |    54.1   | 61.9 | 66.7 |      45.8     | 51.3 |
> |   Ours  | 89.4 |   93.3   | 94.9 | 95.3 | 66.1 |    72.9   | 75.8 | 76.5 |      57.3     | 60.7 |
>
> >***W4: Formal issues***
>
> Thank you for your suggestion regarding the format. Hypotheses (1) and (2) will be more appropriately expressed as research questions; we will fix "optimal" to "better" in the main text.
>
> **References:**
>
> [1] A closer look at memorization in deep networks. ICML 2017.
>
> [2] Teddy: Efficient large-scale dataset distillation via taylor-approximated matching. ECCV 2024.

---

> > ### Comment · Reviewer_B86A · 2025-08-02
> >
> > Thank you for your answer. Answers to Q1 and Q3 are clear and well-addressed.
> > However, you do not conclude on the impact of hyperparameters M and N tuning time on AT-BPTT's overall adoptability.

---

> > > ### Author Response · Authors · 2025-08-02
> > > **Thank you for your new suggestion**
> > >
> > > Thank you for acknowledging our reply and providing new suggestion. We will add the following conclusion to our answer: Based on our practical strategies and consumption assessments, the overhead of low- and medium-sensitivity hyperparameters is negligible, so the table in Computational Cost of Hyperparameter Tuning represents almost the cost of tuning M and N. Moreover, the time and memory overhead of tuning M and N is minimal. For large-scale high-resolution datasets like ImageNet-1K, tuning M and N requires under 80 GPU-hours and 44 GB of memory. Consequently, this tuning has a minimal impact on AT-BPTT's overall adoptability. In future work, we will explore more automatic methods to further eliminate hyperparameter burden.

---

### Official Review · Reviewer_zMmc · 2025-06-29

**Clarity:** 4
**Significance:** 3
**Originality:** 3
**Rating:** 5
**Confidence:** 3

**Summary:**

This paper proposes AT-BPTT, a novel framework to enhance inner-loop optimization in dataset distillation. The key motivation stems from the observation that deep neural networks exhibit different learning behaviors at early, middle, and late training stages, making existing random or fixed truncation strategies suboptimal. AT-BPTT introduces a dynamic truncation strategy guided by gradient magnitude and training stage, an adaptive window size based on gradient variation to retain important updates, a low-rank Hessian approximation to reduce computation and memory costs, and a patch-wise semantic preservation module to enable effective distillation on high-resolution datasets.

**Questions:**

See weaknesses.

**Ethical Concerns:**

["NO or VERY MINOR ethics concerns only"]

**Final Justification:**

I have carefully read the authors' response as well as the comments from other reviewers. The detailed content has adequately addressed my concerns. I believe this is a strong paper in the field of dataset distillation and will maintain my original score of “Accept.” While the performance on ImageNet-1K is slightly below that of diffusion-based methods, the superior results on smaller-scale datasets are convincing. I recommend that the authors include these additional results in the revised version.

**Limitations:**

Yes

**Quality:**

4

**Strengths And Weaknesses:**

Overall, I did not identify any major weaknesses. This is a strong and well-executed paper with solid technical contributions and extensive empirical validation. While the core ideas extend prior work (e.g., RaT-BPTT) in a relatively incremental manner, the proposed framework integrates multiple innovations (e.g., dynamic truncation, adaptive window sizing, and low-rank Hessian approximation). The empirical results are compelling, and the method demonstrates strong scalability and practical efficiency across diverse datasets, including high-resolution benchmarks.

Minor Weaknesses:
1. The paper does not provide a detailed theoretical justification or ablation for how the three training stages (early, middle, late) are precisely defined or separated beyond empirical heuristics.

2. The adaptive window size relies on gradient variation, but its parameterization (e.g., range of adjustment d) may require manual tuning per dataset; clearer guidelines or automated adaptation would improve practical usability.

3. The method introduces several new hyperparameters (e.g., stage transition thresholds, temperature for softmax), which could limit its deployment in real-world settings without additional tuning efforts.

4. The paper lacks comparison with recent SOTA generative dataset distillation methods (e.g., Minimax [1], IGD [2]).

5. While image classification results are strong, it would strengthen the paper to show effectiveness on other downstream tasks such as neural architecture search and continual learning.

[1] Efficient Dataset Distillation via Minimax Diffusion

[2] Influence-Guided Diffusion for Dataset Distillation

---

> ### Author Rebuttal · Authors · 2025-07-30
>
> Thank you for highlighting our solid technical contributions, integrated innovations, extensive validation, and scalable efficiency across diverse datasets. Below we will address specific questions.
>
> > ***W1: Justification and ablation of three stages***
>
> Thank you for your suggestion regarding spliting the learning process into three stages. This is a question that interests Reviewers aCEq, zMmc and B86A. We address it with a Justification and Ablation Study:
>
> ### Justification:
> Based on the theory in paper [1] that is cited as [1] in the main text (early stages prioritize simple patterns and later stages refine complex features), we initially divided the stages into two parts (early and late). However, such improvements in results are limited, as shown in the the table below. This inspires us to explore a more refined staged strategy. Additionally, according to paper [2] that is cited as [9] in the main text, overfocusing the truncted window on a certain stage is not conducive to global learning. Therefore, it is crucial to retain a period of random windowing to cover the entire trajectory. Together, a three-stage strategy is needed. According to the analysis of **Figures 1** and **3** in the main text, this extra stage is defined as the middle stage, and finally boosts performance.
>
> In addition, why not more stages, such as four stages? First, each extra stage adds two new hyperparameters in the stage transition, which complicates tuning and deployment; Moreover, according to the analysis in **Figure 1** in the main text, middle-stage performance demonstrates negligible sensitivity to truncation position (lines 46). A finer partitioning thus produces negligible change. Therefore, we ultimately chose to use a three-stage strategy.
>
> ### Ablation Study:
>
> We conduct ablation experiments on three variants to verify the effectiveness of three-stage stragety. Using CIFAR-10 with IPC=10 (600 epochs) as an example, the setups are as follows:
>
> - **Two-stage variant:** The training is split into two stages of 300 epochs each. In the first stage, truncations position follow $ P_{trunc}(t) $ for early stage in **Equation 6**. In the second stage, truncation positions follow $ P_{trunc}(t) $ for late stage in **Equation 6**.
> - **Three-stage variant:** The training is split into three stages of 200 epochs each. The first and third stages are identical to the first and second stages of the *two-stage variant*, respectively. The second stage employs random truncation.
> - **Four-stage variant:** The training is split into four stages of 150 epochs each. The strategies for the first and third stages align with the first and second stages of the *two-stage variant*, while the second and fourth stages employ random truncation.
>
> The experimental results are shown in the table below:
>
> | Two-stage variant | Three-stage variant | Four-stage variant |
> | :---------------: | :-----------------: | :----------------: |
> |     69.6±0.2      |      70.5±0.3       |      70.3±0.2      |
>
>
> Table above demonstrates that the three-stage variant consistently outperforms both other two variants.
>
> In conclusion, both Justification and Ablation Study support our choice of a three-stage design as an effective and balanced truncation strategy.
>
>
> > ***W2/W3: Hyperparameter fine-tuning***
>
> Thank you and Review B86A for your suggestions regarding hyperparameter fine-tuning. Based on your comments and Reviewer B86A's W2/Q2, we provide following comprehensive reply.
>
> We agree that clarifying the hyperparameter sensitivity, computational overhead, and practical strategies for setting these parameters is crucial for reproducibility and real-world deployment. Therefore, we provide clear guidelines to minimize tuning effort. Our analysis divides these hyperparameters into three categories based on dataset sensitivity, and we discuss empirical basis, practical strategy and computational cost.
>
>
> ### Sensitivity Category
> - Low sensitivity: Fixed defaults suffice across datasets due to universal gradient dynamics
> - Medium sensitivity: Adjusted via simple scaling rules, minimizing need for per-dataset tuning
> - High sensitivity: Only M and N require dataset-specific tuning
>
> ### Empirical Basis and Practical Strategy
> |Hyperparameter|Sensitivity|Empirical Basis|Practical Strategy|
> |:-:|:-:|:-:|:-:|
> |$\tau$ (temperature for dynamic truncation and adaptive window)|Low Sensitivity|Gradient magnitude distributions are similar across dataset, which makes $\tau$ less sensitive to dataset. We conduct ablation study from CIFAR-10 to ImageNet-1K with $\tau$ = 1.0 without re-tuning. Accuracy varied <±0.5% when $\tau$ is perturbed ±0.5 on CIFAR-10/ImageNet-1K.|$\tau$ = 1.0 for all datasets.|
> |k_min (LRHA min rank)|Low Sensitivity|k_min is invariant to dataset. k_min = 1 worked for all tested datasets. Higher k_min will add unnecessary cost.|k_min = 1 for all datasets.|
> |k_max scale (LRHA max rank factor)|Low Sensitivity|Fixed scaling factor.|Not a tunable hyperparameter.|
> |$d$ (window adjustment range)|Medium Sensitivity|Window size exhibits a direct correlation with the total number of timesteps $T$. $T$ is related to dataset scale. Larger datasets require larger windows to capture complex dynamics. |$d \in [10,50]$ for the scale of CIFAR-10 to ImageNet-1K|
> |X, Y (transition counters)|Medium Sensitivity|X and Y depend on training duration. We thus set X, Y proportionally to total epochs.|X = 15% of total epochs; Y = 12.5% of total epochs.|
> |M, N (accuracy variation thresholds)|High Sensitivity|M and N depend on convergence speed of dataset. Different datasets require different thresholds: e.g., M=1.5 for CIFAR-10, but M=1.0 for Tiny-ImageNet due to slower early convergence.|Ablation study in **Stage Transition Threshold Analysis** (lines 295-304).|
>
> ### Computational Cost of Hyperparameter Tuning
>
> The computational cost for tuning AT-BPTT hyperparameters is low, as most parameters are low-sensitivity or medium-sensitivity. As shown in table above, the costs of low-sensitivity or medium-sensitivity hyperparameters are negligible. The hyperparameter tuning is thus focused on M and N. And this process for other datasets is similar to that of ablation study in **Figure 5**. The computational costs on Nvidia A800 are shown in table below:
>
> | Dataset |CIFAR-10  | CIFAR-100| Tiny-ImageNet | ImageNet-1K|
> |:-:|:-:|:-:|:-:|:-:|
> |Time (h) | <25 | <27 | <40 | <80 |
> |Memory (GB) | <7 | <9 | < 18 | < 44 |
>
> If the paper is accepted, we will provide the analysis above in the camera-ready version.
>
>
> > ***W4: Comparison with generative DD methods***
>
> Thank you for your suggestion regarding the comparison with SOTA generative DD methods. We provide the experimental results in the table below:
>
> | Dataset |      | CIFAR-10 |      |      | CIFAR-100 |      |      | Tiny-ImageNet |      |     ImageNet-1K||
>  |:-:|:-:|:-:|:-:|:-:|:-:|:-:|:-:|:-:|:-:|:-:|:-:|
>  | IPC     | 1    | 10       | 50   | 1    | 10        | 50   | 1    | 10            | 50   |1|10|
>   | Minimax [3]     | -    | -    | -| -    |-     | - | - | -         |- |-|44.3|
>   | IGD  [4]   | -    | -        | 66.8 | -    | 45.8      | 53.9 | -    | -             | -    |-|46.2|
>   | Ours    | 54.4 | 72.4     | 78.7 | 36.9 | 49.0      | 55.9 | 24.3 | 32.7          | 48.7 |14.7|30.6|
>
> Through comprehensive comparison, we get several conclusions:
> - The results on ImageNet-1K of Minimax and IGD are both 44.3 and 46.2, which are higher than our results.
> - These two generative methods can not generalize across datasets with various resolutions like CIFAR-10, CIFAR-100, and Tiny-ImageNet, while our AT-BPTT achieves comparable results across these datasets, especially in low-resolution datasets.
> - These two generative methods necessitate the incorporation of diffusion models for distillation, thereby incurring substantial GPU memory consumption and additional computational overhead. In contrast, our method with a LRHA module achieves a $3.9 \times$ training speedup and 63% memory savings compared to the baseline.
> - As shown in **Table 8** in **Appendix C.5**, our AT-BPTT extends the scale of synthetic dataset to high-IPC cases and achieves expressive performance.
>
> If the paper is accepted, we will add the new comparison table to the camera-ready version.
>
>
> > ***W5: Application of downstream tasks***
>
> Thank you for highlighting the importance of downstream evaluations. As stated in **Appendix F Discussion**, we outline several downstream tasks as our future work, like neural architecture search and federated learning. To initially verify the effectiveness of our method for downstream tasks, we have extended language tasks in **Appendix C.8** and pretrained model fine-tuning in **Appendix C.9**. For instance, the results shown in **Figure 7** demonstrate that pre-training on data distilled by AT-BPTT significantly outperforms random selection. These experiments demonstrate that our method preserves critical representations that are transferable to complex downstream tasks beyond simple classification. Additionally, we further discuss AT‑BPTT’s theoretical benefits for diverse applications in **Appendix F Discussion**. Therefore, we will continue to explore AT-BPTT to other downstream tasks in future work.
>
> **References:**
>
> [1] A closer look at memorization in deep networks. ICML 2017.
>
> [2] Embarrassingly simple dataset distillation. ICLR 2024.
>
> [3] Efficient Dataset Distillation via Minimax Diffusion. CVPR 2024.
>
> [4] Influence-Guided Diffusion for Dataset Distillation. ICLR 2025.

---

> > ### Comment · Reviewer_zMmc · 2025-08-01
> >
> > I have carefully read the authors' response as well as the comments from other reviewers. The detailed content has adequately addressed my concerns. I believe this is a strong paper in the field of dataset distillation and will maintain my original score of “Accept.” While the performance on ImageNet-1K is slightly below that of diffusion-based methods, the superior results on smaller-scale datasets are convincing. I recommend that the authors include these additional results in the revised version.

---

> > > ### Author Response · Authors · 2025-08-01
> > > **Thank you for your valuable suggestions**
> > >
> > > Thank you for your continued support and insightful recommendation, which has helped us to further improve our manuscript. In the final version, we will include additional experiments. We appreciate your valuable suggestions and look forward to enhancing the paper’s completeness and impact.

---

### Official Review · Reviewer_KQDB · 2025-07-03

**Clarity:** 3
**Significance:** 4
**Originality:** 4
**Rating:** 5
**Confidence:** 4

**Summary:**

This paper proposes Automatic Truncated Backpropagation through Time (AT-BPTT) to improve upon previous bilevel optimization methods by accounting for the training dynamics of both the inner loop and the outer loop.  To account for inner loop learning dynamics (i.e. learning the parameters of the ConvNet on the distilled data), the truncation window is determined probabilistically by utilizing the inner gradient magnitude and variations. The exact formulation of the probabilistic mechanism is depends on the training stage of the outer loop (i.e. learning the distilled data). Additionally, to reduce the memory overhead, a low Hessian approximation is used. Results on CIFAR-10, CIFAR-100, Tiny-ImageNet, and ImageNet-1K shows SOTA dataset distillation performance.

**Questions:**

1. How is the distilled data initialized?
2. How does it perform in large IPC settings (50 pic) for ImageNet-1K?
3. What does the distilled ImageNet data look like?
4. How well does this method work using ResNets as the model for dataset distillation? These models tends to be more difficult due to the noisy gradients within the inner loop.

**Ethical Concerns:**

["NO or VERY MINOR ethics concerns only"]

**Final Justification:**

Authors answered my confusions and will maintain my score.

**Limitations:**

yes

**Quality:**

4

**Strengths And Weaknesses:**

Strength
1. Addressing the noisy nature of the inner loop in the bilevel optimization problem for dataset distillation is important and critically understudied. This paper provides some much needed insights on the training dynamics of bilevel optimized distilled data.
2. The proposed algorithm AT-BPTT uses a novel idea of dynamically adjusting the inner loop behavior based of the magnitude and variation of inner loop gradients
3. Low-rank Hessian approximation and Patch-wise Semantic Preservation are important contributions towards making dataset distillation less resource intensive
4. The proposed AT-BPTT achieves SOTA performances in CIFAR-10, CIFAR-100, Tiny ImageNet, and ImageNet-1K
5. The ablation studies and evaluation is comprehensive. The paper show that the strategy for selecting window size and location as well as the patch-wise semantic preservations all contribute to the final improvement. Cross architecture generalization and high IPC experiments are also included in the appendix.
6. Code is anonymized and provided

Weaknesses and typos
1. The presentation of the algorithm itself should be included in the main paper instead of appendix and can be improved: outline in Algorithm 1 is a bit confusing-the inner loop should be more explicitly written
2. Missing BPTT citation in the context of dataset distillation [1].
3. Segmennt typo in Algorithm 1

[1] Deng, Zhiwei, and Olga Russakovsky. "Remember the past: Distilling datasets into addressable memories for neural networks." Advances in Neural Information Processing Systems 35 (2022): 34391-34404.

---

> ### Author Rebuttal · Authors · 2025-07-30
>
> Thank you for acknowledging our novel dynamic adjustment, resource‑saving Hessian and semantic preservation contributions, SOTA results across benchmarks, and comprehensive evaluations with available code. Below we will address specific questions.
>
>
> > ***W1/W2/W3: Presentation of algorithm, missing citation and typo***
>
> Thank you for your suggestions regarding the presentation of algorithm, missing citation and typo. If the paper is accepted, we will adjust the layout to place the algorithm in the main text; We will clearly indicate the positions of the inner and outer loops using comments; We will add the BPTT reference; We will correct the typo of "Segmennt" to "Segment".
>
>
> > ***Q1: Data initialization***
>
> We have reported the data initialization procedure in **Appendix B** (lines 507-509). Specifically, We draw synthetic images from a standard Gaussian distribution and assign labels so that each class has an equal number of samples. Following this random generation, we apply a norm-normalization step to unit length. This initialization creates balanced, model‑agnostic starting data that is comparable across experiments.
>
>
> > ***Q2: Performance of high-IPC setting***
>
> We appreciate your interest in the high‑IPC settings on ImageNet-1K. We have completed this experiment during the rebuttal period. With IPC=50, AT-BPTT achieves a test accuracy of 46.5% on ImageNet-1K. This accuracy is substantially higher than the 30.6% at the IPC=10 setting. It provides strong evidence that our method's performance scales robustly with an increasing IPC, rather than being effective only at low-IPC settings. Additionally, we have reported the results on other three datasets (CIFAR-10/100, Tiny-ImageNet) with high-IPC settings in **Appendix C.5** and **Table 8**.
>
> > ***Q3: Data visualization***
>
> While the rebuttal system does not support displaying figures, we will include a detailed ImageNet-1K data visualization in the final version, including:
> - We will present a side‑by‑side grid of $5 \times 2$ distilled samples alongside Teddy [1] and other high‑resolution baselines.
> - Compared to low‑resolution distilled datasets, our ImageNet‑1K distillates exhibit sharper, more recognizable object outlines.
> - The backgrounds become washed‑out or texture‑like, since the model integrates more class features.
>
>
>
> > ***Q4: Performance on ResNets***
>
> We have conducted experiments using the ResNet-18 architecture in **Appendix C.4** and **C.5**.
>
> - In **Appendix C.4** (lines 557-564), we have evaluated the distilled CIFAR-10 dataset with IPC=10 and 50 across four distinct architectures, including ResNet-18. As shown in **Table 7**, the performance of our method on ResNet-18 is lower than on ConvNet, which is the same as the judgement you mentioned. In this case, our AT-BPTT still outperforms all baselines, achieving the current SOTA accuracy of 75.6% under the IPC=50 setting. This provides strong evidence that AT-BPTT has excellent cross-architecture generalization.
>
> - In **Appendix C.5** (lines 565-578), we have further conducted experiments with high-IPC settings, consistently using ResNet-18. The results in **Table 8** demonstrate that AT-BPTT comprehensively surpasses all baselines, across CIFAR-10/100 and Tiny-ImageNet.
>
> In conclusion, these findings confirm AT‑BPTT’s robust cross‑architecture generalization, even on ResNets.
>
> **References:**
>
> [1] Teddy: Efficient large-scale dataset distillation via taylor-approximated matching. ECCV 2024.

---

> > ### Comment · Reviewer_KQDB · 2025-08-04
> >
> > Thank you for the reply. I have read the rebuttal as well as other reviews. I believe this paper provides important contributions to the field of dataset distillation and will maintain my score.

---

> > > ### Author Response · Authors · 2025-08-04
> > > **Thank you for your valuable suggestions**
> > >
> > > Thank you for acknowledging our answer. We sincerely appreciate your valuable suggestions, which will greatly help us improve the paper.

---

### Official Review · Reviewer_aCEq · 2025-07-03

**Clarity:** 3
**Significance:** 3
**Originality:** 4
**Rating:** 5
**Confidence:** 4

**Summary:**

This paper advances Rat-BPTT, by introducing an automatic approach to determine the alignment range. The authors propose a dynamic truncation position selector that uses gradient magnitudes to probabilistically choose where to start backpropagation, and also useing adaptive window sizing to adjusts the truncation window length. The method is further extended to high-resolution images with a patch-wise distillation strategy. Extensive experiments show that AT-BPTT achieves SOTA results among inner-loop methods and is highly competitive with other methods across benchmarks.

**Questions:**

1. Lack justification and ablation of split the learning process into three stages. What if we only define two stages, or more stages? Some ablation experiments would help.
2. What is the actual lengths of the three stages for each experiment?
3. What is the approximation accuracy of LRHA? Will the errors affect overall distillation performance?
4. Why is the PSP module not working well with small resolution images.

**Ethical Concerns:**

["NO or VERY MINOR ethics concerns only"]

**Final Justification:**

I appreciate the authors' feedback and my concerns are well addressed. The additional ablation is a strong complement to the paper. I would raise my rating to "Accept".

**Limitations:**

yes

**Paper Formatting Concerns:**

- Equation 5 and 6 use duplicated notation P_trunc, potentially leading to confusion.

**Quality:**

3

**Strengths And Weaknesses:**

## Strengths

- The paper is well motivated. The authors find the core obstacle in RBTT paradigm, and propose automatic  solution to enhance the current random truncation solution.
- The method achieves substantial improvements in both accuracy and computational efficiency.
- The paper is very clearly written with sufficient implementation details.

## Weaknesses


My main concern is the lack of justification to the  algorithm design. The motivation is sound, but the authors' solution is complex and some design choice is empirical. But overall, I tend towards acceptance due to the good motivation and performance.

---

> ### Author Rebuttal · Authors · 2025-07-30
>
> Thank you for appreciating our clear motivation, detailed presentation, and the significant accuracy and efficiency gains of our method. Below we will address specific questions.
>
> > ***W1: Lack of justification to the algorithm design***
>
> Thank you for recognizing our strong motivation and gains. Based on Q1, we infer that your concerns come from the three-stage strategy, which we will address in the next section.
>
> > ***Q1: Justification and ablation of three stages***
>
> Thank you for your suggestion regarding spliting the learning process into three stages. This is a question that interests Reviewers aCEq, zMmc and B86A. We address it with a Justification and Ablation Study:
>
> ### Justification:
> Based on the theory in paper [1] that is cited as [1] in the main text (early stages prioritize simple patterns and later stages refine complex features), we initially divided the stages into two parts (early and late). However, such improvements in results are limited, as shown in the the table below. This inspires us to explore a more refined staged strategy. Additionally, according to paper [2] that is cited as [9] in the main text, overfocusing the truncted window on a certain stage is not conducive to global learning. Therefore, it is crucial to retain a period of random windowing to cover the entire trajectory. Together, a three-stage strategy is needed. According to the analysis of **Figures 1** and **3** in the main text, this extra stage is defined as the middle stage, and finally boosts performance.
>
> In addition, why not more stages, such as four stages? First, each extra stage adds two new hyperparameters in the stage transition, which complicates tuning and deployment; Moreover, according to the analysis in **Figure 1** in the main text, middle-stage performance demonstrates negligible sensitivity to truncation position (lines 46). A finer partitioning thus produces negligible change. Therefore, we ultimately chose to use a three-stage strategy.
>
> ### Ablation Study:
>
> We conduct ablation experiments on three variants to verify the effectiveness of three-stage stragety. Using CIFAR-10 with IPC=10 (600 epochs) as an example, the setups are as follows:
>
> - **Two-stage variant:** The training is split into two stages of 300 epochs each. In the first stage, truncations position follow $ P_{trunc}(t) $ for early stage in **Equation 6**. In the second stage, truncation positions follow $ P_{trunc}(t) $ for late stage in **Equation 6**.
> - **Three-stage variant:** The training is split into three stages of 200 epochs each. The first and third stages are identical to the first and second stages of the *two-stage variant*, respectively. The second stage employs random truncation.
> - **Four-stage variant:** The training is split into four stages of 150 epochs each. The strategies for the first and third stages align with the first and second stages of the *two-stage variant*, while the second and fourth stages employ random truncation.
>
> The experimental results are shown in the table below:
>
> | Two-stage variant | Three-stage variant | Four-stage variant |
> | :---------------: | :-----------------: | :----------------: |
> |     69.6±0.2      |      70.5±0.3       |      70.3±0.2      |
>
>
> Table above demonstrates that the three-stage variant consistently outperforms both other two variants.
>
> In conclusion, both Justification and Ablation Study support our choice of a three-stage design as an effective and balanced truncation strategy.
>
> > ***Q2: Lengths of the three stages***
>
> Table below reports the average number of epochs in each stage for IPC = 10 across our four benchmarks.
>
> | Dataset | CIFAR-10 | CIFAR-100 | Tiny-ImageNet | ImageNet-1K |
> | :-----: | :------: | :-------: | :-----------: | :---------: |
> |   *IPC*   |    10    |    10     |      10       |     10      |
> |  Early  |   149    |    185    |      244      |     417     |
> | Middle  |   233    |    252    |      405      |     320     |
> |  Late   |   218    |    163    |      351      |     263     |
>
> As detailed in **Threshold-guided Stage Transition** mechanism (lines 195-207), the transitions between stages are triggered automatically by the model's performance dynamics during training. Therefore, each stage’s duration naturally adapts to the dataset’s learning dynamics under a given IPC.
>
>
> > ***Q3: Impact of LRHA***
>
> We have evaluated the impact of our LRHA in **Appendix C.2** (lines 538-544). As shown in **Table 5**, we test the experimental results before and after the introduction of LRHA, using CIFAR-10 IPC=10 as an example. After the introduction of LRHA, the accuracy is reduced from 72.7% to 72.4%, with a negligible drop of just 0.3 %. In contrast, LRHA cuts peak memory usage by 63.3 % and speeds up training by  $3.9 \times$. These results demonstrate that LRHA introduces minimal approximation error while delivering substantial computational savings. Additionally, the experimental results reported in the main text are the results after applying LRHA.
>
> > ***Q4: Performance of PSP***
>
> We appreciate your interest in the behavior shown in **Table 2**. In fact, PSP performs better on high-resolution datasets, while PSP has only negligible improvements on low-resolution datasets. This discrepancy stems from low-resolution patches containing too little semantic content for PSP to exploit.
>
> For low-resolution images (e.g., CIFAR-10, $32 \times 32$), PSP results in each patch being only $8 \times 8$ pixels in size. An $8 \times 8$ pixel block contains limited semantic information, making it difficult for the model to learn meaningful representations. Therefore, for datasets like CIFAR-10 or CIFAR-100, distilling the complete $32 \times 32$ image directly is a much more effective strategy. For high-resolution images (ImageNet-1K, $224 \times 224$), following the same $4 \times 4$ division rule, a $224 \times 224$ image is segmented into 4 patches, each of size $56 \times 56$ pixels. A $56 \times 56$ patch is large enough to preserve rich local structure critical for distillation. Therefore, PSP is a specialized component designed to tackle the challenges of distilling high-resolution images.
>
>
> >***Paper Formatting Concerns:***
>
> Thank you for your suggestion on paper formatting. We will fix the $ P_{trunc} $ in **Equation 6** to $P*_{trunc}$.
>
> **References:**
>
> [1] A closer look at memorization in deep networks. ICML 2017.
>
> [2] Embarrassingly simple dataset distillation. ICLR 2024.

---

> > ### Comment · Reviewer_aCEq · 2025-08-02
> >
> > I appreciate the authors' feedback and my concerns are well addressed. The additional ablation is a strong complement to the paper.

---

> > > ### Author Response · Authors · 2025-08-02
> > > **Thank you for your valuable suggestions**
> > >
> > > Thank you for acknowledging our answer. We sincerely appreciate your valuable suggestions, which will greatly help us improve our paper.

---

### Note · Authors · 2025-08-12

We appreciate all reviewers for their thorough evaluation and insightful feedback on our paper. We summarize the recognized strengths from reviewers' comments. We are pleased that most concerns are addressed, including two critical ones. Based on the feedback, we will incorporate several key improvements in the final version.

### Recognized Strengths
- **Novel Motivation:** The novel approach is "well motivated" (Reviewer aCEq), "provides some much needed insights" (Reviewer KQDB), and has "strong and well-executed" technical contributions and integration of "multiple innovations" (Reviewer zMmc).
- **Impressive Performance:** AT-BPTT achieves substantial improvements in both accuracy and computational efficiency across multiple benchmarks (Reviewer aCEq, KQDB, zMmc, and B86A).
- **Comprehensive Evaluation:** The paper is "very clearly written with sufficient implementation details" (Reviewer aCEq), and praised for its "comprehensive experiments" (Reviewer KQDB), "extensive empirical validation" (Reviewer zMmc).

### Addressing Key Concerns
- **Justification for the Three-Stage Strategy:** A primary concern (Reviewer aCEq, zMmc, and B86A) is the justification for our three-stage training paradigm. This design is grounded in both theory and empirical validation. Theoretically, it combines the insights from "DNNs learn simple patterns first" and "overfocusing on specific stages is suboptimal", necessitating a random-windowing middle stage. Empirically, the results confirm that the three-stage strategy provides better balance.
- **Hyperparameter Practicality:** The second concern is about the practical strategies of hyperparameters (Reviewer zMmc and B86A). We have thus provided a comprehensive analysis, categorizing all hyperparameters based on their sensitivity. We demonstrate that most parameters only need fixed defaults or simple scaling rules. We show that the computational cost for tuning these two high-sensitivity parameters is minimal and has a negligible impact on AT-BPTT's overall adoptability.

### Commitment to Revisions
We will add the new experiments as described in our response. We will also include the justification and the hyperparameter sensitivity analysis. Additionally, we will clarify key definitions, correct all typos, and include new visualizations. We believe that these revisions will strengthen the paper and more clearly articulate the practicality of our contributions.

---

### Decision · Program_Chairs · 2025-09-17

**Decision:**

Accept (poster)

**Comment:**

This paper introduces AT-BPTT, a novel framework for efficient backpropagation featuring several key innovations: a probabilistic stage-aware timestep selector, an adaptive window size strategy driven by gradient variation, a low-rank Hessian approximation (LRHA) for computational efficiency, and a patch-wise semantic preservation (PSP) module for handling high-resolution data. Comprehensive experiments on multiple image datasets demonstrate that AT-BPTT achieves a significant average accuracy improvement of over 6%, alongside a 3.9x speedup in optimization and a 63% reduction in memory cost.

The reviewers' primary concerns centered on the motivation for the proposed three-stage splitting strategy and the transferability of the method's hyperparameters across different models and datasets. The authors' rebuttal effectively addressed these points by providing additional ablation studies and a detailed cross-dataset analysis, which satisfied all reviewers. Further concerns regarding comparisons with state-of-the-art data distillation algorithms, the individual impact of the LRHA and PSP components, and the role of high Images-Per-Class (IPC) were also adequately clarified.

Given the persuasive author response, all reviewers have updated their scores to reflect a strong consensus for acceptance. This paper presents a method of considerable technical novelty, demonstrated through rigorous empirical validation and shown to have broad applicability. The AC strongly recommends acceptance.